# Construction of an Exudative Age-Related Macular Degeneration Diagnostic and Therapeutic Molecular Network Using Multi-Layer Network Analysis, a Fuzzy Logic Model, and Deep Learning Techniques: Are Retinal and Brain Neurodegenerative Disorders Related?

**DOI:** 10.3390/ph16111555

**Published:** 2023-11-02

**Authors:** Hamid Latifi-Navid, Amir Barzegar Behrooz, Saleh Jamehdor, Maliheh Davari, Masoud Latifinavid, Narges Zolfaghari, Somayeh Piroozmand, Sepideh Taghizadeh, Mahsa Bourbour, Golnaz Shemshaki, Saeid Latifi-Navid, Seyed Shahriar Arab, Zahra-Soheila Soheili, Hamid Ahmadieh, Nader Sheibani

**Affiliations:** 1Department of Molecular Medicine, National Institute of Genetic Engineering and Biotechnology, Tehran 1497716316, Iran; h_latifi@nigeb.ac.ir (H.L.-N.); davarimlh@gmail.com (M.D.); nargeszolfaghari90@yahoo.com (N.Z.); s.piroozmand@gmail.com (S.P.); staghiz3@uwo.ca (S.T.); zssoheili@gmail.com (Z.-S.S.); 2Departments of Ophthalmology and Visual Sciences and Cell and Regenerative Biology, University of Wisconsin School of Medicine and Public Health, Madison, WI 53705, USA; 3Department of Human Anatomy and Cell Science, University of Manitoba College of Medicine, Winnipeg, MB R3T 2N2, Canada; am.barzegar.behrooz@gmail.com; 4Electrophysiology Research Center, Neuroscience Institute, Tehran University of Medical Sciences, Tehran 1416634793, Iran; 5Department of Virology, Faculty of Medicine, Hamadan University of Medical Sciences, Hamadan 6517838636, Iran; salehjamehdor24@gmail.com; 6Department of Mechatronic Engineering, University of Turkish Aeronautical Association, 06790 Ankara, Turkey; mlatifinavid@thk.edu.tr; 7Department of Physiology and Pharmacology, Schulich School of Medicine & Dentistry, Western University, London, ON N6A 5C1, Canada; 8Department of Biotechnology, Alzahra University, Tehran 1993893973, Iran; m.bourbour1@yahoo.com; 9Department of Studies in Zoology, University of Mysore, Manasagangothri, Mysore 570005, India; golnazshemshaki110@gmail.com; 10Department of Biology, Faculty of Sciences, University of Mohaghegh Ardabili, Ardabil 5619911367, Iran; s_latifi@uma.ac.ir; 11Biophysics Department, Faculty of Biological Sciences, Tarbiat Modares University, Tehran 1411713116, Iran; shahriar.arab@gmail.com; 12Ophthalmic Research Center, Research Institute for Ophthalmology and Vision Science, Shahid Beheshti University of Medical Sciences, Tehran 1666673111, Iran; hahmadieh@gmail.com

**Keywords:** age-related macular degeneration, deep learning, diabetic retinopathy, fuzzy logic, multi-layer network, neurodegenerative disorders

## Abstract

Neovascular age-related macular degeneration (nAMD) is a leading cause of irreversible visual impairment in the elderly. The current management of nAMD is limited and involves regular intravitreal administration of anti-vascular endothelial growth factor (anti-VEGF). However, the effectiveness of these treatments is limited by overlapping and compensatory pathways leading to unresponsiveness to anti-VEGF treatments in a significant portion of nAMD patients. Therefore, a system view of pathways involved in pathophysiology of nAMD will have significant clinical value. The aim of this study was to identify proteins, miRNAs, long non-coding RNAs (lncRNAs), various metabolites, and single-nucleotide polymorphisms (SNPs) with a significant role in the pathogenesis of nAMD. To accomplish this goal, we conducted a multi-layer network analysis, which identified 30 key genes, six miRNAs, and four lncRNAs. We also found three key metabolites that are common with AMD, Alzheimer’s disease (AD) and schizophrenia. Moreover, we identified nine key SNPs and their related genes that are common among AMD, AD, schizophrenia, multiple sclerosis (MS), and Parkinson’s disease (PD). Thus, our findings suggest that there exists a connection between nAMD and the aforementioned neurodegenerative disorders. In addition, our study also demonstrates the effectiveness of using artificial intelligence, specifically the LSTM network, a fuzzy logic model, and genetic algorithms, to identify important metabolites in complex metabolic pathways to open new avenues for the design and/or repurposing of drugs for nAMD treatment.

## 1. Introduction

Formation of new blood vessels from pre-existing capillaries is essential for the pathogenesis of many diseases with a neovascular component, such as cancer, proliferative diabetic retinopathy (PDR), and exudative or neovascular age-related macular degeneration (nAMD) [1,2,3]. nAMD is a leading cause of irreversible visual impairment in older adults [4]. Due to the major role of the vascular endothelial growth factor (VEGF) and VEGF receptor 2 (VEGFR2) signaling pathway in driving angiogenesis, several antibody-based and tyrosine kinase inhibitory systems have been designed to limit VEGF-VEGFR2 interactions and/or interfere with the downstream signaling pathways [5]. Despite promising results with anti-VEGF monotherapies in different neovascular diseases, a plethora of current studies reveal an incomplete response to anti-angiogenic drugs (including anti-VEGF) in a significant portion of nAMD patients [6,7].

Several mechanisms are under investigation for the development of new anti-angiogenic drugs to overcome the lack of response to anti-VEGF. These include the compensatory angiogenic pathways, vessel co-option, intussusceptive microvascular growth, and vascular mimicry [8,9,10,11,12,13]. In addition, recent studies demonstrate important crosstalk among angiogenesis signaling pathways and other factors involved in different biological processes [10]. Thus, a broader investigation of angiogenesis signaling pathways and their interactions with other biological processes are essential for a holistic approach to better understand the angiogenesis phenomena and its association with other biological processes and to identify novel targets to overcome the extended lack of response [14,15,16]. To accomplish this goal, the utilization of a biological network analysis approach is essential [17,18,19,20].

Network-based approaches can be used to study complex biological diseases and integrate multiple types of data. Disease-related genes tend to be arranged in clusters as disease modules. A disease module represents an interconnected set of mechanisms that are linked to a phenotype. Protein–protein interaction (PPI) network analysis is used to comprehensively investigate complex intracellular signaling pathways and identify disease-related genes. Indeed, topological analysis of the networks for evaluating different kinds of centralities and identification of hubs as highly connected nodes play fundamental roles in distinguishing novel targets and avoiding inefficient responses [20,21]. A gene regulatory network (GRN) is a set of genes that interact to regulate the activation of a specific cell function. Gene regulation activities are perturbed or malfunctional in many diseases. Consequently, reconstructing this type of network is applicable to studying some hub gene regulation processes.

Here, we integrated key proteins that are involved in 12 related angiogenic signaling pathways. These included VEGF-VEGFR2, FGF-FGFR, EGF-EGFR, Dll4-Notch, TGFβ-ALK1, HGF/c-Met, angiopoietins/Tie receptors, Wnt/β-catenin, PDGF/PDGFR, Ephs/Ephrins, IGF-IGFR, factors related to vessel co-option, intussusceptive microvascular growth-related proteins, vascular mimicry-related factors, text mining data [22], angiogenesis-related protein–protein interaction networks (the extended angiome; first neighbors are linked to proteins in the extended angiome, and factors are not linked to any proteins in the angiome or extended angiome) [23], angiogenesis-related inflammatory factors, endothelial cell (EC) metabolism-related genes, endoplasmic reticulum stress-related factors, angiogenesis-related immune checkpoints, autophagy signaling pathways, cytoskeleton remodeling factors, wound response, neurogenesis, vision-related genes, aging-related factors, vitamin D-related signaling pathways, G-protein-coupled receptor signaling pathways, prostaglandin signaling pathways, and 87 signaling pathways that are directly or indirectly linked to angiogenesis processes to reconstruct a comprehensive angiogenesis-related PPI network. Furthermore, recent studies and multiple databases of disease-related target identification were integrated to reconstruct an exhaustive nAMD network [24].

## 2. Results

### 2.1. The NeDRex Plugin’s Network for Identifying Disease Modules

The nAMD-related modules and their components were identified using the NeDRex plugin through the utilization of two algorithms (MuST and DIAMOnD) (Figure 1). The SQSTM1, C3, and RPGR genes were the center of three disease modules discovered by the MuST algorithm. The DIAMOnD algorithm’s findings included both isolated vertices and nodes that are viewed as parts of a network. According to the analysis of the network portion, the DIAMOnD algorithm detected 11 modules centered on the genes TIMP3, APOE, CFB, SQSTM1, CFH, C3, TLR4, VEGsssFA, C2, CFI, and TNFRSF10A. As a result of the aforementioned algorithms, the following schematic figure was produced (Figure 2). Two distinct strategies were developed to evaluate the genes derived from the algorithms. The centrality analysis of a comprehensive network containing all genes identified using these two algorithms and the hubs discovered by analyzing AV-DRN and AMD-PPIN revealed 31 important genes. As a preliminary step, these 31 genes were utilized in construction of the gene regulatory network (Figure 3). The second method was used to conduct a detailed investigation of nAMD disease. Thirteen genes that significantly affected the disease were identified as common key items between two algorithms (yellow nodes). The nodes related to AMD disease in Figure 1 appear in blue, but when identifying disease modules in Figure 2, they are changed to yellow due to the plugin’s default settings.

### 2.2. Gene Regulatory Network Analysis

Six miRNAs (hsa-mir-124-3p, hsa-mir-335-5p, hsa-mir-661, hsa-mir-29b-3p, hsa-mir-29c-3p, and hsa-mir-450a-1-3p) and fourteen essential genes were found when the miRNA-gene regulatory network was constructed using the aforesaid thirty-one genes as input data. Indeed, the 14 essential genes at this stage were the same genes that were related to miRNAs in the presented gene regulatory network (Figure 3A). In the following steps, the findings of the constructed lncRNA-miRNA interaction network indicated that four lncRNAs (NEAT1, KCNQ1OT1, SNHG17, and XIST) interacted with four of the six identified miRNAs (hsa-mir-124-3p, hsa-mir-335-5p, hsa-mir-29b-3p, and hsa-mir-29c-3p), and they appear to be crucial in controlling these miRNAs’ function (Figure 3B).

### 2.3. Enrichment Analysis

Three different layers of nAMD-related data (genes, miRNAs, and metabolites) were subjected to the enrichment process. The results of enrichment on 31 identified genes showed that seven pathways involving Staphylococcus aureus infection (false discovery rate (FDR) = 0.000152), protein digestion and absorption (FDR = 0.00031), ECM–receptor interaction (FDR = 0.00427), amoebiasis (FDR = 0.00492), the AGE-RAGE signaling pathway in diabetes complications (FDR = 0.00492), focal adhesion (FDR = 0.00492), and complement and coagulation cascades (FDR = 0.0401) represent significant influences. According to the miRNA enrichment findings, all six discovered miRNAs play a role in the first five pathways, which are significant in terms of FDR. These pathways included terpenoid backbone biosynthesis (FDR = 0.0022855), arachidonic acid metabolism (FDR = 0.0280283), the hippo signaling pathway—multiple species (FDR = 0.0280283), base excision repair (FDR = 0.0311627), and complement and coagulation cascades (FDR = 0.0311627). Applying two databases (KEGG and SMPDB) and 115 extracted metabolites as input data, metabolite enrichment analysis was carried out. Nine significant pathways were identified using the KEGG–metabolite enrichment analysis in terms of FDR. These included aminoacyl–tRNA biosynthesis (FDR = 7.02 × 10^−12^); glyoxylate and dicarboxylate metabolism (FDR = 0.000176); arginine biosynthesis (FDR = 0.00562); alanine, aspartate, and glutamate metabolism (FDR = 0.0204); sphingolipid metabolism (FDR = 0.027); glycine, serine, and threonine metabolism (FDR = 0.0288); cysteine and methionine metabolism (FDR = 0.0288); valine, leucine, and isoleucine biosynthesis (0.0354); and taurine and hypotaurine metabolism (FDR = 0.0354). Furthermore, the SMPDB–metabolite enrichment analysis revealed two distinct pathways involving the urea cycle (FDR = 0.0308) and glycine and serine metabolism (FDR = 0.0458). Schematic figures of the metabolite enrichment results are presented in Figure 4.

The false discovery rate (FDR) index, along with the KEGG and SMPDB databases, was utilized to identify significant pathways. The SMPDB database revealed two important pathways: the urea cycle (FDR = 0.0308) and glycine and serine metabolism (FDR = 0.0458). However, nine significant pathways were identified based on the KEGG database, including aminoacyl–tRNA biosynthesis (FDR = 7.02 × 10^−12^); glyoxylate and dicarboxylate metabolism (FDR = 0.000176); arginine biosynthesis (FDR = 0.00562); alanine, aspartate, and glutamate metabolism (FDR = 0.0204); sphingolipid metabolism (FDR = 0.027); glycine, serine, and threonine Metabolism (FDR = 0.0288); cysteine and methionine metabolism (FDR = 0.0288); valine, leucine, and isoleucine biosynthesis (FDR = 0.0354); and taurine and hypotaurine Metabolism (FDR = 0.0354).

### 2.4. Metabolite Pathway Analysis

The integration of the metabolic pathway topology analysis with the enrichment analysis of metabolic pathways is also crucial. Two different sorts of results were produced, depending on whether relative betweenness centrality (R-b C) or out-degree centrality (O-d C) was applied in the topological analysis. Figure 5A shows the criteria that were considered for this analysis. Two criteria were used to examine and summarize the results. First, we chose the top five cases in each of the two centralities (relative betweenness centrality (R-b C) and out-degree centrality (O-d C)) based on their impact parameter values. Second, an assortment of the results based on FDR was conducted, and three cases (phenylalanine, tyrosine, and tryptophan biosynthesis; synthesis and degradation of ketone bodies; and D-glutamine and D-glutamate metabolism) were removed since they had no significance in the metabolic pathway analysis (red color). These analyses found four significant pathways, including taurine and hypotaurine metabolism (FDR = 0.03605); alanine, aspartate, and glutamate metabolism (FDR = 0.021047); glycine, serine, and threonine metabolism (FDR = 0.029711); and aminoacyl–tRNA biosynthesis (FDR = 8.02 × 10^−12^).

### 2.5. Joint Pathway Analysis

We conducted a joint pathway analysis to examine the connection between 115 distinct AMD-related metabolites obtained from published articles along with 31 crucial genes involved in these metabolic pathways. Three different sorts of findings were obtained depending on the chosen topology measure (degree, betweenness, or closeness; Figure 5B). Two criteria were used to examine and summarize the results. The first step involved choosing the top five cases in each of the centralities based on their impact parameter scores. The results were classified based on FDR in the following step, and three cases (synthesis and degradation of ketone bodies, the citric acid cycle (TCA cycle), and glycolysis or gluconeogenesis) were excluded because they were not significant in the joint pathway analysis. As a consequence, the six items identified as critical metabolic related pathways included alanine, aspartate, and glutamate metabolism (FDR = 0.0005232); glycine, serine, and threonine metabolism (FDR = 0.000031504); arginine biosynthesis (FDR = 0.00083089); sphingolipid metabolism (FDR = 0.0072172); cysteine and methionine metabolism (FDR = 0.000016837); and arginine biosynthesis (FDR = 0.000831).

### 2.6. Metabolite–Gene–Disease Interaction Network

A broad perspective of putative functional connections among metabolites, associated genes, and the target diseases was provided by the metabolite–gene–disease interaction network. The outcomes additionally demonstrated a connection between some altered metabolites in nAMD and other neurodegenerative disorders. For instance, it was found that there was an association between schizophrenia and eight metabolites. These included L-lactic acid, cortisol, cholesterol, (R)-3-hydroxybutyric acid, glycine, L-lysine, L-arginine, and pyruvic acid. Additionally, a connection was found between Alzheimer’s disease (AD) and four metabolites: glycine, L-lysine, L-arginine, and calcium (Figure 6A).

### 2.7. Identification of Genes Related to nAMD–Single Nucleotide Polymorphisms (SNPs)

The two-centrality metrics (degree and betweenness) in each of the two drawn networks (the SNP–Gene–Disease network and the SNP–Gene–Metabolite–Disease network) were used as input data in the mGWAS-Explorer database to identify 30 significant SNPs associated with nAMD disease. The genes to which these SNPs connect were first identified and included in the third list of crucial genes in nAMD. The second phase involved specifying shared SNPs between nAMD, Alzheimer’s disease (AD), multiple sclerosis (MS), Parkinson’s disease (PD), and schizophrenia. In a comparison between the 30 critical SNPs in nAMD with the total number of shared SNPs in nAMD and the four other neurodegenerative disorders (15 SNPs), 9 SNPs were found to be essential for the centrality parameters in nAMD disease and to be associated with a range of aforementioned neurodegenerative disorders. The main results were categorized into three axes: (i) the rs1061170 (corresponding to the complement factor H (CFH) gene) was common in nAMD, AD, MS, and schizophrenia; (ii) the rs699947 (corresponding to the VEGFA gene) was common in nAMD, AD, MS, PD, and schizophrenia; and (iii) the rs429358 (corresponding to the APOE gene) was common in nAMD, AD, MS, and PD (Figure 6B).

### 2.8. Results of the Developed Binary-GA Search Method for Maximization of the Model in the 56-Dimensional Space

The maximization progress using 300 iterations of the binary-GA is shown in Figure 7. The binary-GA method generates a chromosome (an array with a length of 55), mimicking the input sequence of metabolites. Each element in the array is represented as either one or zero, indicating its effectiveness in influencing the output merit of the metabolic route. A value of one denotes a highly impactful metabolite, while a value of zero implies a metabolite with a lesser influence on the overall output. Through rigorous application of the binary-GA optimization technique, we successfully identified the 25 most valuable metabolites within the metabolic pathway. These metabolites exhibited a significant effect on the output merit, contributing to the overall functionality and regulation of the biological process under investigation. The selected metabolites are listed in Table 1.

### 2.9. Summary of All Results

To acquire a clear understanding of the nAMD condition, all the results were pooled in four tables. Table 2 lists three significant gene lists, six miRNAs, four lncRNAs, and seven metabolites that were shared between the MMIN and MGDIN networks via degree and betweenness centralities as well as twenty-five metabolites based on the fuzzy logic model, deep learning, and the genetic algorithm. Table 3 lists the common metabolites between AMD, schizophrenia, and AD; 30 AMD-SNPs; and common SNPs between AMD and AD, MS, PD, and schizophrenia. Table 4 summarizes the findings of the pathway enrichment analysis at various levels (including 31 essential genes, six miRNAs, 115 metabolites), the pathway analysis, and the joint pathway analysis. The results that were commonly observed among the metabolic pathway enrichment analysis, pathway analysis, and joint pathway analysis are presented in Table 5.

## 3. Discussion

The diagnostic and therapeutic molecular networks constructed for the nAMD pathogenesis consisted of 30 proteins. These were created by merging 18 genes from the disease modules, 14 genes from the gene regulatory network, and 6 genes from the AMD-SNP data, which were validated. The 30 identified genes were in three general families, including angiogenesis, inflammation, and metabolism. A detailed examination of the angiogenesis processes revealed six different subfamilies. These included the angiogenesis itself (VEGFA, PDGFA, and MUC1), ECM proteins (Col1A1, Col1A2, Col4A1, Col14A1, Col18A1, TIMP3, and P3H3), cytoskeleton remodeling (FN1, C1QTNF5, and PVRL2), protein-mediated transport (ABCA4), proteasome degradation (UBC), cell viability (TNFRSF10A), and DNA repair (ERCC6). The subfamilies related to inflammation included the complement system (C1S, C2, C3, C9, CFB, CFI, and CFH), proteins involved in toll-like receptors (TLR4), the autophagy process (SQSTM1), and protein involved in PINK1-PRKN mediated mitophagy (TOM40). Finally, the metabolism subfamily was classified into three subfamilies, which included glucose metabolism (SLC16A8), lipoprotein metabolism (APOE), and NAD^+^ metabolism (NMNAT1). Recent studies have also shown that the complement system plays a crucial role in AMD pathogenesis through regulation of ECM stability, inflammation, energy metabolism, lipid accumulation, and oxidative stress (OxS). Understanding the ECM’s structural components is important before considering how the complement system affects their function.

### 3.1. ECM Proteins, Complement System, and Pathogenesis of nAMD

ECM molecules play a crucial role in modulating cellular functions and various processes, including angiogenesis. Collagen, laminin, and fibronectin are examples of ECM molecules that exhibit angioregulatory characteristics [25]. Our findings also demonstrated that functional members exist among each of the four key classifications, including Col1A1 and Col1A2 belonging to the fibril-forming collagens, Col4A1 belonging to the basement membrane collagens and network-forming collagens, Col14A1 belonging to fibril-associated collagens with interrupted triple helices, and Col18A1 as a candidate from multiplexin subgroups [26,27]. COL1A1 expression affects cell migration, survival, and recurrence in diabetic retinopathy (DR) and malignant astrocytoma patients. Additionally, the decreased level of COL1A1 significantly limits the expression of many proteins linked to cell invasion, including STAT3, matrix metalloproteinase 2 (MMP2), MMP9, and NF-κB [28].

A 150 bp DNA element in the promoter region of the COL1A2 gene is a TGFβ-responsive element. TGFβ promotes the expression of PDGFA, fibronectin (FN1), and collagens (COL1A1 and COL4A1), with a prominent role in angiogenesis [29,30,31]. Also, circular COL1A2 (circCOL1A2) controls the miR-29b/VEGF axis throughout the pathological progression of DR. Since the miR-29b/VEGF axis is not specific to DR, targeting circCOL1A2 may be a possible therapeutic for DR and other retinal diseases with a neovascular component [32]. COL4A1 manages the inhibition of cell death; the activation of focal adhesion kinase; the cell cycle; tumor angiogenesis; the PI3K/MAPK, PI3K/AKT, and PRL/PAK1 signaling pathways, and the discoidin domain receptor (DDR) axes. Collagen 4 increases the production of CCL7 protein via the PI3K/MAPK pathway, which in turn triggers epithelial mesenchymal transition (EMT) and metastasis. MMP2/9 secretion, migration, invasion, and colonization of tumor cells are stimulated by COL4A1 through activation of the PI3K/AKT signaling pathway and DDRs. Additionally, the network-forming collagens, such as COL4A1, may be involved in the control of the IGFR signaling pathways, apoptosis, and autophagy. This may be due to the interactions between DDRs and IGFR and the function of DDRs in regulating Bcl-2, Bcl-xl, Survivin, Bax, and LC3II expression. MMP1-3 secretion in breast cancer cells is stimulated by the activation of the PRL/PAK1 signaling pathway via COL4A1.

The Collagen type XIV α 1 chain (COL14A1) reveals its key roles in various processes such as ECM and collagen fibril organization. Moreover, collagen binding results showed that high expression of COL14A1 is associated with poor overall survival rates for breast cancer patients [33]. Collagen XVIII (COL18A1) is another significant collagen with essential roles in the maintenance of retinal structure and neural tube closure [34,35]. Prolyl 3-hydroxylase genes (P3H1, P3H2, and P3H3), which are crucial for proper post-translational modifications of the collagen chains and creation and maturation of their final quaternary structures, are also significant components. These genes induce type IV collagen hydroxylation and inhibit its control of platelet aggregation [36]. In addition, the epigenetic inhibition of P3H2 and P3H3 may play a key role in the progression and metastasis of breast cancer [37]. By reducing the expression of genes in this family in some cancers, the possibility of greater interaction of collagen type IV with platelets occurs, protecting cancer cells from shear stress and natural killer cells, thus helping them escape immune system detection [38,39].

Early stages of nAMD are associated with changes in Bruch’s membrane (BrM) and choriocapillaris ECM composition, which can be followed by BlamD, BlinD, and drusen development. These changes most likely produce a situation in which the underlying genetic risk is manifested. ECM protection against C3b deposition and inflammation is an inevitable process [40]. To better understand C3b function, three stages of production, deposition, and regulation of its activity should be considered. All three complement activation pathways (lectin, classical, and alternative) converge in the formation of a protein complex (the C3 convertase), which cleaves C3 into the anaphylatoxin C3a and the central protein in the complement amplification loop (C3b). Multiple genetic and molecular studies have shown that overactivation of the alternative complement system plays a significant role in the pathogenesis of AMD [41]. Six of the identified complement system genes related to AMD diagnostic and therapeutic panel (C2, C3, C9, CFB, CFI, and CFH) belong to the alternative pathway, and one (C1s) belongs to the classical pathway. Thus, inhibition of multiple complement activation pathways should be considered for an effective treatment strategy for AMD.

Infectious endophthalmitis or sterile uveitis/endophthalmitis are two types of problems that are related to intravitreal anti-VEGF injections. These are caused by pathogens or patient/delivery/medication-specific factors, respectively [42,43]. The antibody-triggered classical complement pathway is initiated when circulating C1 complexes are recruited to antibody-labeled pathogen surfaces [44]. C1q, C1r, and C1s are the three major units that constitute the C1 complex. C1q comprises the antibody recognition unit and its associated proteases C1r and C1s, which are activated to cleave other complement proteins that together form enzymes on the surface that catalyze the covalent deposition of C3b molecules onto the bacterial surface [45]. One of the pathways obtained from the KEGG enrichment analysis of the identified important genes was Staphylococcus aureus infection. It seems that the importance of the classical complement pathway (along with the alternative pathway) depends on the C1s-driven immune response to the Staphylococcus aureus infection and the role of Staphylococcus aureus toxins in ocular damage and inflammation [46,47].

### 3.2. ECM Remodeling by Cytoskeleton-Related Proteins and Angioinflammatory Factors

ECM proteins like collagen can be assembled using fibronectin, a key component of the newly deposited ECM, as a template. The hallmark of late-stage DR and nAMD is neovascularization, which uses ECM production as a scaffold for the abnormal new vessel architecture. The fibronectin matrix and collagen turnover are impacted by changes in MMP activity that are crucial in the etiology of nAMD. PDGF and TGFβ contribute to proliferative vitreoretinopathy (PVR), nAMD, and PDR and are known to promote the formation of the fibronectin matrix [48]. The ECM’s main connective protein, fibronectin, is essential for cell migration, adhesion, proliferation, and ERK activation [49,50]. Also, fibronectin associates with collagen type IV [49]. The heparin-II domain of fibronectin binds VEGFA, and the extra domain A of fibronectin promotes VEGFC expression and lymph angiogenesis via the PI3K/AKT signaling pathway [50,51]. In both healthy and pathological situations, fibronectin structure and function are inversely correlated. The fluctuations between the ECM’s two conformations (relaxed and strained) ensure the process of maintaining proper tissue homeostasis. When the ECM is relaxed, fibronectin is frequently seen in a compact or extended form, but when the ECM is strained, fibronectin changes conformation to accommodate the tensile forces generated by the cells. Three axes represent the outcome of these conformational changes: (i) association of soluble substances such as VEGFA with fibronectin; (ii) ECM rearrangements by making it easier for the matrix elements to bind; and (iii) modification and activation of integrins [52].

Mechanical stress in RPE cells also increases gene expression in the axis of angiogenesis (VEGF, ANG2, and HIF-1α(, inflammation (IL6, IL8, and TNF-α) and ECM (FN1, VIM, and CDH2) and has a prominent role in promoting aberrant angiogenesis in nAMD [53]. The process of Tie2/integrin complex stabilization is mediated through fibronectin [54]. Fibronectin is essential for controlling the angiogenesis process and signaling pathways mediated by VEGFR2 and the receptor for advanced glycation end products (RAGE). Under physiological conditions, fibronectin binds VEGF to induce angiogenesis, recruitment of c-Src to VEGFR2, and downstream activities. The glycosylated form of fibronectin, by directly binding to RAGE, leads to greater interaction of c-Src with RAGE and prevents the activation of the VEGF-VEGFR2 signaling pathway [55].

The role of tissue inhibitor of metalloproteinase 3 (TIMP3) in a variety of processes such as angiogenesis, inflammation, ECM remodeling, and amyloid precursor protein (APP) processing has been substantiated. TIMP3 mitigates angiogenesis by inhibiting VEGF, modulates ADAM 10 and ADAM 17 activity in APP processing [56,57,58,59], fine-tunes ADAM 17 activity in inflammation [60,61,62], and affects ECM remodeling by inhibiting MMP2 and MMP9 [63,64]. Reduced transcription (via siRNA or the T allele of rs13278062) or knockout of the tumor necrosis factor receptor superfamily 10A (TNFRSF10A) gene in RPE cells leads to decreased cell viability and increased apoptosis through downregulation of the PKCA pathway. Moreover, OxS and the *Tnfrsf10* null mutation in mice upregulate *TNFRSF10A* transcription and create age-dependent RPE abnormalities [65]. Recent studies have also identified a crosstalk between VEGF and mucin 1 (Muc1). In addition, tumor cells promote different signaling pathways that are involved in growth, survival, and EMT by increasing Muc1 expression on their surface [66]. In hypoxic conditions, mucin 1 stimulates expression of several proangiogenic factors such as HIF-1α, VEGFA, PDGFB, and connective tissue growth factor (CTGF) [66,67]. In addition, by connecting to the MUC1 promoter, HIF-1α increases its expression [68].

The C1q tumor necrosis factor-related protein-5 (C1QTNF5) mutation (S163R) plays a fundamental role in the formation of late-onset retinal degeneration [69]. C1QTNF5 belongs to the C1q/TNF family, which is involved in immunity and inflammation, glucose and lipid metabolism, and vascular maintenance [70,71]. Moreover, the LASSO regression model suggests that C1QTNF5 may be a key biomarker in PDR [72]. Cytoskeleton-remodeling-related proteins play important roles in controlling angiogenesis and vascular permeability. In addition to cadherins, nectins constitute other adhesion molecules localizing to cell–cell junctions [73]. Recent in vitro studies suggest that PVRL2 (CD112 or nectin-2) regulates human EC migration and proliferation. In CD112-deficient mice, the blood vessel coverage of the retina was significantly enhanced. A blockade of CD112 modulated EC migration and significantly enhanced tube formation [74]. Thus, changes in ECM composition and function are key modulators of the cellular microenvironment that impact various cellular pathologies and diseases.

### 3.3. Aging and Pathogenesis of nAMD

Choroid becomes less flexible and thinner with age [75], diminishing the retinal blood flow and nutrient/oxygen supply [76] and thus leading to nutrient and oxygen starvation of the retina (ischemia) and placing the RPE cells under significant metabolic stress. To counteract this ischemic stress, RPE cells drive choroidal neovascularization (CNV), a characteristic of nAMD accounting for about 90% of cases of severe vision loss. VEGF and VEGFRs play fundamental roles in the onset and progression of nAMD [77].

Clinical evidence reveals that the effectiveness of intravitreal injections of anti-VEGF agents such as ranibizumab, bevacizumab, and aflibercept is constrained by competing and compensating alternative angiogenic pathways that could act as escape routes [78,79]. Results from our earlier network analysis indicated that angiopoietin 2 (ANG2) could play a significant role in an incomplete response to anti-VEGF [20,80]. By extending the data examined here, the potential function of the PDGFA protein as the second pro-angiogenic factor involved in anti-VEGF resistance was also predicted. According to an assessment of the disease’s active clinical trials, research is often focused on nine broad categories. An analysis of treatment strategies that focused on two or more angiogenesis-related axes revealed that VEGFA in combination with ANG2 and PDGFA may be more desirable targets for an effective treatment strategy [81].

To counteract the metabolic remodeling, RPE cells may also change their metabolism. Targeting numerous angiogenic pathways has the potential to prevent revascularization, but it also provides other strategies to prevent an incomplete response via altering the cellular metabolism. The cells in the hypoxic and avascular area cooperate together with the cells in the normoxic and nearby vascular region as part of this approach. The metabolism of the cells in the hypoxic region shifts toward increased glycolysis and lactate generation. Nevertheless, the cells near the blood vessels utilize the resulting lactate for oxidative phosphorylation [82,83]. A process called metabolic symbiosis may also be important in the anti-VEGF resistance in the retina [84]. Monocarboxylate (MC) transporters are one way to transport lactate into the cell [85,86]. MCT3 (SLC16A8) is present only in the RPE cells and the choroid plexus endothelium, and MCT4 (SLC16A3) is expressed in greater amounts under hypoxic conditions [87,88]. By oxidizing lactate within the cell, LDHB causes an increase in pyruvate, which enables PDH to inhibit proline hydroxylation. The activity of prolyl hydroxylase leads to poly-ubiquitination and degradation of HIF-1α in the proteasome. Thus, inhibition of prolyl hydroxylase by pyruvate increases the HIF-1α protein stabilization and activation of the NF-κB transcription factor. HIF-1α and NF-κB increase VEGFA and IL8 transcription, respectively [86]. Hence, one possible way to improve the effectiveness of anti-VEGF drugs is to prevent lactate from entering the cells through inhibition of MCT3 (SLC16A8) and MCT4 (SLC16A3).

In addition to genes related to glucose metabolism, those involved in NAD^+^ metabolism (NMNAT1) also are significant to the development of AMD disease. Over the past decade, it has become clear that NAD^+^ impairment plays a significant role in almost all retinal neurodegenerative disorders [89]. Genome sequencing of carriers with Leber congenital amaurosis 9 (LCA9) has revealed approximately 10 mutations in the NMNAT1 gene [90]. The NMNAT1 gene encodes an important enzyme with a key role in nicotinamide adenine dinucleotide (NAD) biosynthesis. Further research on LCA9 retinas has shown that NMNAT1 is not only responsible for NAD^+^ biosynthesis but also is necessary for the development, structure, and function of the retina [91,92]. The malfunction of NMNAT1 can result in harm to the photoreceptors’ survival and vision function, enhance RPE damage caused by ROS, and notably elevate the levels of molecules that trigger inflammation and drusen formation [91,93,94]. Inhibiting the enzymatic activity of NMNAT1 could cause damage to photoreceptors, the outer and inner nuclear layer, and the plexiform layer, ultimately resulting in RPE harm [95,96]. Targeting NMNAT1 via specific shRNA in retinal explants had three general consequences: (i) increased H3 and H4 acetylation levels in the retina; (ii) increased expression of two pro-apoptotic genes (Noxa and Fas); and (iii) an increased number of apoptotic retinal progenitor cells [92].

Three elements comprising radiation exposure, a high fatty acid ratio, and high oxygen consumption increase the retina’s vulnerability to OxS [97]. OxS is a major contributor to the aging process, CNV, progressive retinal degeneration such as AMD, immune cell infiltration, and atrophy [98,99]. The excision repair cross-complementing 6 (ERCC6) gene plays a vital role in the aging process, transcription-coupled nucleotide excision repair, and the quick removal of RNA polymerase II-blocking lesions from the transcribed strand of active genes. It also plays a significant role in complex formation at DNA repair sites and in ocular degeneration [100,101]. AMD susceptibility is increased as a result of decreased ERCC6 expression, its mutation, and synergistic interaction with CFH mutations [100,101,102,103,104,105].

### 3.4. AMD and Other Neurodegenerative Diseases

Drusen formation, immune system activation, and retinal inflammation are the three criteria used to show how apolipoprotein metabolism, the complement system, and amyloid beta (Aβ) interact to cause AMD. Drusen is formed between RPE and photoreceptors. Subretinal drusenoid deposit (SDD) or reticular pseudodrusen (RPD) are between RPE and Bruch’s membrane as basal linear deposits (BLinD) or soft drusen [106,107]. BLinD is more involved in nAMD, while RPD is a major factor in disease progression toward geographic atrophy [108,109]. Overall, the drusen contents are classified into five general categories: lipids, apolipoproteins, complement factors, minerals, and other proteins. Phospholipids, triglycerides, cholesterol, apolipoprotein E (APOE), CFH, and vitronectin are common between BLinD and RPD [110]. RPE and Müller glial cells produce APOE, a plasma lipid transport protein. APOE is secreted from the apical and basal surfaces of RPE cells. Moreover, APOE is regarded as a transporter of cholesterol and lipids in the development of drusen and plays a crucial function in lipid efflux and trafficking from BrM to choriocapillaris [111,112]. The APOE disruption causes impairments in retinal function by thickening the retina and altering the lamina’s elasticity [113].

Exposure of human RPE cells to serum C1q leads to the formation of APOE-, Aβ-, and vitronectin-rich sub-RPE deposits and shows the intercommunication between complement factors and APOE [114]. An interaction between Aβ and APOE has also been recognized. RPE and retinal ganglion cells express Aβ and APP, which are important in the development of ocular aging, AMD, and AD [115,116,117,118]. APOE isoforms can affect Aβ in a wide range of biological processes and molecular functions such as oligomer stabilization [119,120], binding and clearance [121,122], APP transcription, and Aβ secretion [123]. However, the pattern of these isoforms and their impact on the retina and brain is distinct. Compared to the typical allelic variation of APOE3 (Cys112 Arg158), APOE2 (Cys112 Cys158) and APOE4 (Arg112 Arg158) represent an inverse effect in the risk of developing AMD and AD [124,125,126]. The alleles APOE2 and APOE4 confer increased and decreased risks of AMD, respectively, except in the Chinese population. Conversely, APOE2 plays a protective role, while APOE4 is the primary genetic risk factor for AD, a condition that shares characteristics with AMD such as neuroinflammation and Aβ deposition [110,127,128]. Studies have also demonstrated a role for Aβ in the development of angiogenesis and inflammation. The six axes—C3b, the membrane attack complex, the presence of CFH in amyloid vesicles, increased expression of CFB in RPE cells, inhibition of CFI, and production of inflammatory cytokines by macrophages and microglia—are involved in the interaction of Aβ with complement factors [116,129,130,131,132,133].

Exposure of RPE cells to Aβ leads to increased IL6, IL8, IL33, and VEGF expression through the AGE-RAGE and TLR4/MyD88/NF-κB signaling pathways; decreased PEDF expression; increased formation of angiogenic tubules in EC; induction of NLRP3 inflammasome formation; cytokine production; and finally cytoskeleton remodeling of RPE cells [134,135,136,137,138]. Thus, there seems to be a close relationship among complement factors, lipoprotein metabolism, Aβ, TLRs, and angiogenic factors. Another classification that is significant in AMD is the TLR4-UBC13-ABCR4 axis. Recognition of damage-related molecular patterns (DAMPs) through pattern-recognition receptors (PRRs) such as TLR4 activates the innate immune system [139,140,141,142,143]. Expression of TLR4 both at the RNA and protein level is observed in retinal EC, RPE cells, Müller glial cells, choroidal EC, and photoreceptors [144,145,146,147,148,149,150,151]. DAMPs binding to TLR4 activate two distinct downstream signaling pathways that are dependent on MyD88 and TRIF proteins. Both of these molecules have similar effects on angiogenesis and inflammation by activating the NF-κB and MAPK signaling pathways [152,153,154]. The MyD88-dependent signaling pathway leads to the formation of the myddosome complex, which consists of MyD88, IRAK1, and IRAK4 proteins. IRAK1/4 and UBC13 together play a significant role in the poly-ubiquitination of K63 in TRAF6 [155,156]. The final consequence of this action is the activation of NF-κB related genes [152]. Recent studies demonstrated that TLR4 signaling pathway generally plays a role in four axes: photo-oxidative stress, cell viability, choroidal neovascularization, and inflammatory pathways [154,157,158,159]. Moreover, genetic variants associated with TLR4 that affect AMD have been identified [160,161,162]. Mice with a defect in ATP binding cassette subfamily A member 4 (Abca4)/retinol dehydrogenase 8 (Rdh8) genes show an AMD-like phenotype and exhibit long-term sensitivity to light. Increased expression of TLR2/4 and a wide range of pro-inflammatory cytokines are the consequences of exposing mice with defects in Abca4/Rdh8 to light [163].

### 3.5. Autophagy and AMD

Another important mechanism involved in AMD is the autophagy (SQSTM1)–mitophagy (TOM40) axis. Defects in lysosomal clearance (a lower rate of autophagy flux) along with increased accumulation of waste substances play a prominent role in the development of AMD [164,165,166]. Interaction between the ubiquitin proteasome system (UPS) and autophagy play a major role in the removal of cellular waste materials [167]. The protein that acts as a bridge between these two processes is SQSTM1 [168]. LC3 has binding sites for ubiquitin and SQSTM1. Recognition of the ubiquitinated cargo by SQSTM1 and its further binding to LC3 leads to the initiation of autolysosomal degradation [169]. Mitochondrial dysfunction, mtDNA damage, and increased ROS production lead to protein aggregation and inflammation in AMD [170,171]. In order to prevent the increased production of ROS, the damaged mitochondria must be removed through the mitophagy process. The PTEN-induced kinase 1 (PINK1), Parkin RBR E3 ubiquitin protein ligase (PRKN), and optineurin (OPTN) proteins play key roles in this process [172]. TOM40 is a translocase that is used to import nascent proteins through the mitochondrial outer membrane and plays a key role in PINK1-PRKN mitophagy, autophagy, aging, and mitochondria–endoplasmic reticulum contact sites [169,173,174,175,176,177,178]. Overexpression of TOM40 in mitochondria leads to caspase-dependent cell death and plays a role in the degeneration of the primarily eye nerve tissue. In recent studies, the role of this protein in the formation of Parkinson’s and late-onset AD (LOAD) has been also demonstrated [179,180].

### 3.6. Non-Coding RNAs and AMD

The miRNA–gene regulatory network analysis revealed six miRNAs, including hsa-miR-661, hsa-miR-29c-3p, hsa-miR-29b-3p, hsa-miR-124-3p, hsa-miR-450a-1-3p, and hsa-miR-335-5p, along with 14 significant genes involved in nAMD and other neovascular retinopathies. We found miR-29b-3P and miR-29c-3P interacting with genes including COL1A1, COL4A1, PDGFA, MUC1, and BMP1. The miR-29 family members could impact some EC functions and neovascularization: miR-29b inhibits angiogenesis and cell proliferation by targeting VEGF and PDGFB in retinal microvascular EC [181], while miR-29c suppresses the migration and angiogenesis of human EC by targeting IGF-1 [182]. Consistent with previous studies, our findings also demonstrated that miR-29c targets collagen gene expression in the retina. The expression of the miR-29 family was reported to significantly decrease in the ECs of patients suffering from Fuchs endothelial corneal dystrophy (FECD) [183]. Thus, overexpression of miR-29b and miR-29c resulted in a considerable downregulation of several ECM genes such as COL1A1, COL4A1, and LAMC1 in corneal EC. These studies suggested that the miR-29 family may affect RPE cells through the regulation of ECM gene expression [183,184].

miR-661 manifests different functions in a cell-specific manner. The expression of miR-661 is upregulated in various cancers (including non-small-cell lung cancer (NSCLC)) and promotes the proliferation, migration, and invasion processes in NSCLC cells [185]. A high expression level of miR-661 is observed in serum from both dry and wet AMD patients. However, a significantly higher expression level of miR-661 (4.7×) was recorded in patients with dry AMD compared to wet AMD [186]. Thus, miR-661 may function in the development of AMD through pathological pathways other than angiogenesis. miR-124 is the most abundant miRNA detected in the central nervous system. In the retina, miR-124 is also highly expressed in photoreceptors and plays a role in both retinal homeostasis and pathological conditions like AMD. An anti-inflammatory role for miR-124-3p in retinal neurons was previously revealed. Accordingly, its dysregulation is documented with the pathogenesis of inflammatory diseases, neurological disorders in the brain, and pathogenesis of AMD in the retina [187]. Chu-Tan et al. (2018) reported that miR-124 directly targets a number of chemokines like C-C motif ligand 2 (CCL2 or MCP-1) that are upregulated in nAMD [188]. Our studies showed that miR-124-3p could target a number of genes, including COL1A1, COL4A1, and MUC1. As described above for miR-29b and miR-29c, miR-124 is similarly postulated to regulate ECM production by targeting COL1A1 and COL4A1. Furthermore, miR-124 acts as a tumor suppressor miRNA by inhibiting multiple genes involved in pathways related to different cancers and degenerative diseases including cell proliferation, apoptosis, angiogenesis, migration, and invasion [189].

Another anti-angiogenic miRNA is miR-450a-1-3p. According to our findings, miR-450a-1-3p targets several genes, including COL1A1, COL4A1, and UBC. Thus, miR-450a-1-3p could control angiogenesis by repressing EC proliferation and migration in nAMD. The last reported anti-angiogenic miRNA is miR-335-5p, which is similarly known as a tumor suppressor that modulates cell proliferation and migration in various cancers [190,191]. Most gene targets of miR-335-5p emerged from miRNA–gene regulatory networks, including BMP1, MUC1 [192], P3H3 [193], FN1 [194], SERPING1 [195], and C1S [196]. Thus, miR-335 is thought to regulate ECM homeostasis and cell proliferation and migration by modulating these genes. Since anti-angiogenic therapy is accepted as a viable therapeutic approach for nAMD, taking advantage of anti-angiogenic miRNAs targeting multiple significant genes may be promising in combination therapies to control angiogenesis pathways in nAMD.

In parallel with the lncRNA-miRNA interaction network, we made a prediction of four lncRNAs, including nuclear-enriched abundant transcript 1 (NEAT1), KCNQ1 opposite strand/antisense transcript 1 (KCNQ1OT1), small nucleolar RNA host gene 17 (SNHG17), and X-inactive specific transcript (XIST), with possible involvement in nAMD and other neovascular retinopathies. Nuclear-enriched abundant transcript 1 (NEAT1) has been extensively studied in choroidal, retinal, and corneal neovascularization. NEAT1 is one of the long non-coding RNAs classified as ocular neovascular LncRNAs and performs its regulatory function through epigenetic mechanisms including DNA methylation, histone methylation, and histone acetylation [197]. miR-194-5p/DNMT3A prevent NEAT1 promoter region methylation (promote NEAT1 expression) and increase cell migration and invasion [198]. Moreover, NEAT1 expression is regulated by the EGFR-STAT3 and NF-κB (p65) axis, and H3K27 trimethylation—via the binding of NEAT1 to EZH2—activates the Wnt/β-catenin signaling [199]. NEAT1 regulates neovascularization by sponging miR-377 and increasing expression of VEGFA, SIRT1, and BCL-XL through a possible histone acetylation mechanism [200]. Microphthalmia-associated transcription factor (MITF) is a key transcription factor in RPE cells. The NEAT1–splicing factor proline- and glutamine-rich (SFPQ)–MITF axis plays a critical role in RPE cell proliferation [201].

Pyroptosis is a kind of programmed cell death that, in terms of appearance and process, varies from apoptosis to autophagy, necroptosis, ferroptosis, and NETosis. Cytokines like IL-1 and IL-18 must be released for pyroptosis. Several ocular illnesses, including AMD, glaucoma, DR, dry eye disease, keratitis, uveitis, and cataract, are linked to pyroptosis [202,203,204,205,206,207]. Development of dry eye disease is significantly influenced by the miR-214-3p–caspase 1 axis and KCNQ1 opposite strand/antisense transcript 1 (KCNQ1OT1) [208]. Additionally, miR-486a-3p inhibition and NLRP3 upregulation are responsible for KCNQ1OT1-induced pyroptosis [209]. Furthermore, KCNQ1OT1 may enhance EMT and angiogenesis via overexpression of RAB11A [210]. Small nucleolar RNA host gene 17 (SNHG17) is another essential lncRNA that plays a crucial part in angiogenesis. SNHG17, a new member of the SNHG family, is significantly expressed in a variety of malignancies and may have carcinogenic properties. Several studies have shown the connection between SNHG17 and the growth, invasion, migration, apoptosis, and drug resistance of tumor cells. Clinical research has linked high SNHG17 expression to a poor prognosis [211]. Colorectal adenocarcinoma cell proliferation and migration are facilitated by SNHG17 through inhibition of miR-23a-3p, which modifies CXCL12-mediated angiogenesis [212]. Hence, SNHG17 acts in angiogenesis (and possibly in nAMD) and deserves further investigation.

The most well-known lncRNA to date is X-inactive specific transcript (XIST) [213]. It was discovered that when high glucose concentration stressed ARPE-19 cells, XIST was downregulated, and the cells showed enhanced apoptosis and reduced migration [214]. By reducing apoptosis and regaining migratory capacity, XIST overexpression shielded ARPE-19 cells from the stress brought on by high glucose levels. In those cells, XIST bound to and inhibited miR-21-5p, indicating that it may serve as a sponge for miRNA-21-5p. These interactions may present the hypothesis that female sex is a potential risk factor for AMD [215]. XIST lncRNA is also essential for angiogenesis. Expression of VEGF signaling in human brain microvascular EC under hypoxic conditions was dependent on XIST, and XIST also plays a critical role in hypoxia-induced angiogenesis via the miR-485-3p/SOX7 axis [216]. Moreover, a mechanistic study showed that by modulating the miR-92a/Itg5 (integrin α5) or KLF4 (Kruppel-like transcription factor 4) axis, the lncRNA XIST may control angiogenesis and reduce cerebral vascular damage after cerebral ischemic stroke, respectively [217].

### 3.7. Metabolic Activity in nAMD and Neurodegeneration

Investigation of the metabolic profiles in an AMD–Alzheimer’s–schizophrenia axis demonstrated that there were three pathways in common: L-glycine, L-arginine, and L-lysine. If L-arginine is metabolized, various products such as nitric oxide (NO) and L-citrulline (via three different isoforms of NO synthase (eNOS/nNOS/iNOS)), agmatine (via arginine decarboxylase), and L-ornithine and urea (via arginase) are produced [218]. Nitric oxide has an indispensable role in a wide range of processes such as synaptic plasticity, neurodevelopment, cerebral blood flow, release of mediators (such as glycine and taurine), neurotoxicity, inflammatory functions, and neurodegeneration [219,220,221,222,223]. Increased NO and NOS expression and decreased arginase activity are noted in people with schizophrenia [224,225,226,227]. In addition, agmatine is considered as a potential schizophrenia-related biomarker [228]. L-arginine and L-lysine compete for connection to the cationic amino acid transporter (CAT). Therefore, L-lysine can be considered a NO synthesis inhibitor [229,230]. Recent studies revealed that betaine is a metabolite that is decreased in the plasma sample of patients with a first episode of schizophrenia and the second sample set. As a consequence, betaine (tri-methyl glycine) could also be considered a biomarker for schizophrenia [231]. The expression of eNOS and nNOS (including as a result NO production) are significantly downregulated in the eyes of patients with AMD [232]. Additionally, iNOS may promote (and downregulation of the iNOS/NO/VEGF signaling pathway may reduce) CNV formation [233,234]. Peroxynitrite (ONOO−), which is produced via the interaction of NO and ROS, also compromises vascular endothelial function [235].

L-lysine plays two potential roles in AMD pathophysiology. First, it works with methionine to create carnitine. Second, it serves as a substrate for mitochondrial electron transport flavoproteins. As a critical characteristic of nAMD, the altered carnitine shuttle and retinal autofluorescence of mitochondrial flavoproteins are found in metabolomics investigations [236,237]. These results demonstrate that three flavoprotein substrates for mitochondrial electron transfer—lysine, proline, and valeryl carnitine—are elevated in patient blood [238]. When triggered by blue light under stress, flavoproteins connect to mitochondrial enzymes in the electron transport chain, oxidize, and emit a green autofluorescence [239,240]. Flavoprotein fluorescence (FPF) can be utilized non-invasively as an indicator of mitochondrial oxidative stress in the retina [241]. Significant FPF elevation is detected in nAMD patients [242]. Discovery of FPF led to a quantifying technique for FPF emission from a patient’s retina, developing imaging tests based on metabolic indicator for predicting disease progression in the retina [243,244,245,246].

The overall angio-regulatory role of glycine has been examined in multiple studies on three contradictory axes. (i) Glycine is a powerful anti-angiogenic nutrient because it activates a glycine-gated chloride channel. By hyperpolarizing the cell membrane, it prevents Ca(2+) inflow and reduces VEGF-mediated signaling [247]. (ii) Glycine stimulates angiogenesis by activating the glycine transporter 1 (GlyT1)–glycine–mTOR–voltage-dependent anion channel 1 (VDAC1) axis [248]. (iii) Glycine affects vascular development in a dose-dependent manner through modulation of VEGF and NOS gene expression. Glycine acts as an anti-angiogenic agent at high concentrations, with a pro-angiogenic effect at low doses [249]. Additionally, a study in a diabetic rat model showed that glycine supplementation reduces retinal neuronal damage [250]. The presence of glycine [251], L-lysine [238,251,252,253], and L-arginine [254] has also been reported in numerous investigations as changed metabolites in AMD.

The L-arginine/NO pathway and related metabolites play a fundamental role in the formation of vascular dementia and AD. Several processes related to neurodegeneration are affected by a lack of NO. These involve reducing synaptic plasticity, brain atrophy and ischemia, activating microglia, severity of dementia, amyloid peptide formation and accumulation, evoking neuroinflammation, and promoting endothelial dysfunction [255,256]. Methylation of L-arginine through class I or II protein arginine methyltransferase leads to the production of asymmetric or symmetric dimethylarginine (ADMA or SDMA), respectively. ADMA ultimately leads to the endogenous synthesis of L-arginine. In people suffering from AD, SDMA and DMA are increased, whereas ADMA, Arg/ADMA, L-arginine, and L-citrulline are decreased [256,257,258].

Glycine also has neuroprotective effects through attenuation of D-galactose (D-gal)-induced oxidative stress. D-gal is an artificial senescence inducer that is used to model brain aging in animals [259]. Moreover, glycine depletion facilitates synaptic dysfunction, apoptotic neurodegeneration, memory impairment, and neuro-inflammatory responses [260]. Oxidative stress activates the c-Jun N-terminal kinase (JNK) signaling pathway and mediates neuroapoptosis [261]. Glycine exhibits its neuroprotective effect by inactivating the JNK signaling pathway [260]. Additionally, systemic or central nervous system investigation of energy metabolism in AD revealed lower concentrations of glycine in plasma and lower concentrations of lysine in plasma and cerebrospinal fluid. Thus, changes in metabolic activities may provide novel indications of pathological changes.

## 4. Materials and Methods

### 4.1. First Data Sources

The Network-based Drug Repurposing and Exploration (NeDRex) platform (https://nedrex.net/ (accessed on 11 February 2019 related to DisGenNET and 6 March 2022 related to OMIM)) was used to identify genes involved in the disease modules for nAMD. This process was undertaken using two algorithms: Multi-Steiner Trees (MuST) and DIseAse MOdule Detection (DIAMOnD). Based on merged results from these two algorithms, the first list of genes involved in nAMD pathogenesis were identified (gene list 1 = 230 genes) (Appendix A) [21]. The second step for target identification and determination of the genes involved in nAMD disease entangled examining related studies and the valid databases such as DisGenNET (https://www.disgenet.org/ (accessed on 4 May 2020)) [262], STRING (https://string-db.org/ (accessed on 17 October 2020)) [263], Disnor (https://disnor.uniroma2.it/ (accessed on 4 January 2018)) [264], the Therapeutic Target Database (TTD) (http://bidd.group/group/cjttd/ (accessed on 28 October 2021)) [265], KEGG disease (accessed on 1 August 2018)) [266,267], and the comparative toxicogenomics database (CTD) (http://ctdbase.org/ (accessed on 17 October 2020)) [268] (gene list 2 = 7061 genes) (Appendix A).

In the third step, various studies were considered, such as those on signaling pathways, protein–protein interaction networks, text-mining-related data, human gene–disease association networks, and microarray data. Several keywords consisting of 14 general headings were used in the process, such as “angiogenesis signaling pathways”, “vessel cooption-related factors”, “vascular mimicry-related factors”, “angiogenesis-related protein-protein interaction networks”, “angiogenesis-related inflammatory factors”, “endothelial cell metabolism”, “endoplasmic reticulum stress”, “angiogenesis-related immune checkpoints”, “autophagy signaling pathways”, “cytoskeleton remodeling factors”, “wound response”, “neurogenesis”, “aging-related factors”, “vision-related genes”, “vitamin D-related signaling pathways”, “G protein coupled receptor signaling pathways”, and “prostaglandin signaling pathways”. We then analyzed the Kyoto Encyclopedia of Genes and Genomes (KEGG) (https://www.genome.jp/kegg/ (accessed on 1 January 2021)) and WikiPathways (https://www.wikipathways.org/index.php/WikiPathways (accessed on 19 November 2020)) databases [269,270] to identify genes involved in 89 signaling pathways directly or indirectly related to anti-VEGF resistance (Figure 8). The results were combined to create three independent lists linked to disease modules of nAMD (gene list 1 = 230 genes), the AMD-related protein–protein interaction network (AMD-PPIN) (gene list 2 = 7061 genes), and the anti-VEGF-resistance-related network (AV-DRN) (gene list 3 = 4340 genes) (Appendix A). The schematic diagram of the process is shown in Figure 8A–D.

### 4.2. Network Construction

To identify nAMD disease modules, we first used the NeDRex plugin (version 1.0.0) implemented in Cytoscape software (version 3.7.2). In the second step, disease modules were identified using two different algorithms: Multi-Steiner Trees (MuST) and Disease Module Detection (DIAMOnD). Reconstruction of comprehensive nAMD and anti-VEGF-resistance-related networks for the Homo sapiens organism was conducted with the GeneMANIA plugin (version 3.5.1) [271] implemented in Cytoscape software (version 3.7.2).

### 4.3. Topological Network Analysis

The CentiScaPe plugin (version 2.2) [272] was used to determine nodes with a high centrality index in both the nAMD (gene list 2) and anti-VEGF-resistance-related (gene list 3) networks individually. Several types of centralities were considered, such as degree, betweenness, centroid value, closeness, bridging, eccentricity, and eigenvector centrality [20] (Appendix A). Nodes with a high centrality index are often referred to as hubs and play crucial roles in networks. According to the results of the two aforementioned networks, we integrated the first 20 genes with the highest scores from each centrality. Following processing of 4340 and 7061 genes, the two gene lists containing 39 and 52 genes were created, respectively (Appendix A). Three criteria were considered when creating the final gene list. These included: (i) The genes associated with the NeDRex platform (230 genes), hub genes determined by analyzing the protein–protein interaction networks involved in nAMD disease (52 genes), and the signaling pathways involved in resistance (39 genes) were integrated together to create a comprehensive gene list (313 genes) (Appendix A). (ii) As part of the centrality analysis, 12 parameters (degree, betweenness, centroid value, closeness, stress, bridging, radiality, eccentricity, eigenvector centrality, clustering coefficient, topological coefficient, and neighborhood connectivity) were considered, and the top five genes from each centrality were chosen (Appendix A). (iii) The identified genes were integrated to yield a final gene list that contained 31 essential genes (Appendix A). Our remaining efforts focused on these genes. A schematic diagram of the procedure carried out is shown in Figure 9.

### 4.4. Gene Regulatory Network Construction

Considering 31 genes as the input data, NetworkAnalyst (https://www.networkanalyst.ca/ (accessed on 15 November 2016)) [273] and miRTarBase v8.0 were applied to create the miRNA–gene regulatory networks [274]. Fourteen genes and six critical microRNAs were found when two centrality criteria (degree and betweenness) were considered. The lncRNA-miRNA interaction network was then created using the miRNet database (https://www.mirnet.ca (accessed on 18 December 2019)) and six miRNAs as the input data [275]. At this point, four indispensable lncRNAs were identified while considering the two aforementioned factors. A schematic diagram of the procedure is represented in Figure 10A.

### 4.5. Second Data Sources: nAMD-Related Metabolites and SNPs

At this point, the procedure of reviewing recent studies, data extraction, and classifications were performed in order to identify metabolites [276,277,278,279] and single-nucleotide polymorphisms (SNPs) [101,280,281,282,283,284,285,286,287,288,289,290,291,292,293] relevant to nAMD disease. As a result, 317 SNPs (Appendix A) and 115 metabolites (Appendix A) that are effective in the development of nAMD were identified. Pathway and joint pathway analyses were used to study the function of these metabolites. The metabolite–gene–disease interaction network was further reconstructed to identify 10 important metabolites via centrality parameters (degree and betweenness) and the potential relationship of the nAMD-related metabolic profile with other neurodegenerative disorders. The MetaboAnalyst 5.0 database (https://www.metaboanalyst.ca/ (accessed on 1 December 2018)) was used for all analyses related to metabolites [294].

Two different types of networks—the SNP–Gene–Disease network and the SNP–Gene–Metabolite–Disease network—were reconstructed to address important SNPs related to nAMD disease via centrality parameters (degree and betweenness) and potential association of the nAMD-related SNP profile with other neurodegenerative disorders. The first 10 cases from each network were chosen based on their degree and betweenness centralities. A list of 30 critical SNPs was created by merging the results (Appendix A). A detailed schematic presentation of these analysis is shown in Figure 10B.

### 4.6. Enrichment Analysis

Functional enrichment analysis was performed at three distinct levels, including genes, miRNAs, and metabolites, using ExpressAnalyst (https://www.expressanalyst.ca/) (accessed on 18 February 2021) [273], microRNA enrichment analysis and annotation (miEAA) (https://www.ccb.uni-saarland.de/mieaa tool/ (accessed on 1 December 2022)) [295], and MetaboAnalyst 5.0 (https://www.metaboanalyst.ca/ (accessed on 9 September 2022)) [294], respectively. The most enriched pathway of the network was determined using the KEGG enrichment analyses. An FDR < 0.05 was considered as representing statistical significance.

### 4.7. Data Pre-Processing for the First Fuzzy Logic Model

Despite identification of a significant number of metabolites based on the network parameters, it remained unclear which metabolite had the most impact on the degree, betweenness, closeness, and FDR. To determine the most significant metabolite, a model was created that considered both the lowest FDR value and the highest impact value after the joint pathway analysis. The subsequent step involved identifying metabolite interactions required for this model. This was accomplished using fuzzy logic (for metabolites and pathways indices) and deep learning techniques. To achieve this goal, specific metabolite indicators and factors related to metabolic pathways underwent two stages of data pre-processing. Initially after the joint pathway analysis, a total of 33 crucial pathways were chosen based on FDR < 0.05 (Appendix A). Subsequently, by transforming CPD codes into compound names, all the metabolites that participated in these pathways were identified (Appendix A). These metabolites were represented by the numbers 0 or 1 to indicate their absence or presence, respectively, in each individual pathway (Appendix A). The subsequent phase involved assessment of the degree and betweenness of each metabolite by reconstructing two different types of networks: (i) a metabolite–metabolite interaction network (MMIN) and (ii) a metabolite–gene–disease interaction network (MGDIN) (Appendix A). The metabolite’s four parameters (degree _MMIN_, betweenness _MMIN,_ degree _MGDIN_, and betweenness _MGDIN_) were normalized within the range of 0 to 1 (Appendix A). Each of the four parameters was analyzed separately to determine the quartiles, and then four categories were established to represent weak (Minimum to Q1), medium (Q1 to Q2), good (Q2 to Q3), and excellent (Q3 to Maximum) effects (Table 6). After evaluating four parameters and their respective impact ranges, a total of 256 rule bases were established, and decisions were made on their consequences (Appendix A). These rules were used to create the first fuzzy logic model, and the number 1, which was previously used to indicate the presence of a metabolite in the metabolic pathway, was replaced by numerical output of the fuzzy logic model obtained for each specific metabolite (Table 7 and Appendix A). Figure 11A displays a schematic representation of the initial stage of the first fuzzy logic model outlining actions that were taken.

### 4.8. Metabolites Merit Calculation Using Fuzzy Logic Model

The first step in our methodology involved calculating a numerical merit for each metabolite. This merit was derived from four independent parameters: two degree values and two betweenness values. The degree values quantified the number of interactions a metabolite had with other metabolites and genes within the network. The betweenness values captured the centrality and influence of a metabolite in mediating interactions between other metabolites and genes. By considering these four parameters, we could comprehensively evaluate the importance of metabolites within the metabolic network. To integrate the four parameters into a single merit, a fuzzy logic model was developed. Fuzzy logic is a potent computational framework that simulates human thinking under ambiguity. It has emerged as a valuable tool in various domains, including decision making, control systems, and pattern recognition. In the context of metabolomics research, fuzzy logic presents a viable method for combining many factors and capturing the intricate interactions seen in metabolite networks. Fuzzy logic makes it possible to represent metabolite properties in a more understandable way and makes it easier to calculate a single merit by using linguistic variables and fuzzy membership functions [296,297,298]. Based on a variety of variables, including interactions between metabolites and between metabolites and genes, this merit represents the overall significance of metabolites.

In this research, the fuzzy logic model took the four parameters as inputs, and each input was represented by four triangular membership functions: Weak, Moderate, Good, and Excellent. These membership functions captured the linguistic values associated with inputs, allowing for a more intuitive representation of the metabolite’s characteristics. The fuzzy logic model processed these inputs and produced an output, termed Merit, which was also defined by four triangular membership functions representing Weak, Moderate, Good, and Excellent values within the range of 0–100. After calculating the numerical merit for all metabolites, the metabolic route database was updated. The calculated merit values were assigned to each metabolite within their respective routes. This process ensured that the importance of metabolites was reflected accurately within the database, providing a reliable foundation for subsequent analyses and evaluations. This process facilitated the development of a standardized and extensible algorithm for evaluating the importance of metabolites within metabolic routes and could be utilized to prioritize metabolites based on their merit values. It facilitated targeted investigations and interventions within metabolic pathways. The general flowchart of the process is shown in Figure 11B. The range of the four triangular membership functions and the applied rule bases are shown in Table 6 and Appendix A. The calculated merits for each metabolite are shown in Table 7.

### 4.9. Data Pre-Processing for the Second Fuzzy Logic Model

The input data for the second fuzzy logic model were derived from the table listing 33 significant pathways from which the outcomes of the first phase of the fuzzy logic model were implemented. During this stage, the impact and FDR of the pathway-related indicators (degree/betweenness/closeness) were analyzed, and their data were sorted into quartiles independently. The quartile results for the three indicators—degree, betweenness, and closeness impact—were established by defining their range as “Minimum to Q1 = D, Q1 to Q2 = C, Q2 to Q3 = B, and Q3 to Maximum = A”. However, the defined range for FDR was determined differently and is presented as “Minimum to Q1 = A, Q1 to Q2 = B, Q2 to Q3 = C, and Q3 to Maximum = D” (Table 8). After evaluating four parameters and their respective impact ranges, a total of 52 rules were established, and decisions were made on their consequences (Appendix A). These rules were used to create the second fuzzy logic model, and the numbers related to four pathway-related indicators were replaced with the numerical output of the fuzzy logic model obtained for each specific pathway (Table 9). Figure 12A displays a schematic representation of the initial stage of the second fuzzy logic model outlining the actions that were taken.

### 4.10. Calculating the Output Merit of the Metabolite Route Using the Fuzzy Logic Model

The second step involved the calculation of the output merit for the metabolic route using fuzzy logic. This step considered four independent parameters that collectively reflected the performance of the metabolic route, namely degree impact, betweenness impact, closeness impact, and FDR. The degree impact, betweenness impact, and closeness impact parameters exhibited a better performance with higher values, while the FDR parameter demonstrated a better performance with a lower value. To integrate these four parameters and calculate a single output merit, a second fuzzy logic model was developed. This model consisted of four inputs and one output. Three inputs, namely degree impact, closeness impact, and FDR, were represented by combined triangular–trapezoidal membership functions, including Weak, Moderate, Good, and Excellent. The second input, betweenness impact, was represented by three combined triangular–trapezoidal membership functions, including Weak, Good, and Excellent, based on the network analysis values. The fuzzy logic model processed these inputs and generated an output termed the Output Merit of Metabolic Route, which was defined by four trapezoidal membership functions representing Weak, Moderate, Good, and Excellent values within the range of 0–100. The overall process flowchart is illustrated in Figure 12B. The range of four independent parameters and the applied rule bases are shown in Table 8 and Appendix A. The calculated merits for each metabolite are shown in Table 9. After updating the database based on calculated output merits for each metabolic route, a database with 33 metabolic routes, 55 metabolites, and one output merit was developed. This database is ready for development of a metabolic route model using a Long Short-Term Memory (LSTM) network and a deep learning technique.

### 4.11. Metabolic Route Model Development Using Long Short-Term Memory (LSTM) Network

The Long Short-Term Memory (LSTM) network, a type of recurrent neural network (RNN) capable of analyzing sequence and time-series data, is a network architecture that is ideally suited for the goal of creating an intelligent metabolic pathway classifier. By using a training set of sequences and goal values, an LSTM neural network can be used to predict a numerical answer within a sequence. An LSTM network processes input data as a recurrent neural network by looping over time steps and updating the network state. The network state includes data that have been stored for earlier time steps. A common LSTM network for regression starts with a sequence input layer and then moves on to an LSTM layer. A fully connected layer and a regression output layer make up the network’s final layers. The general architecture of an LSTM network is shown in Figure 13A.

To construct an LSTM network for sequence-to-one regression, a layer array was created containing a sequence input layer, an LSTM layer, a fully connected layer, and a regression output layer [299,300]. A sequence input layer was employed with an input size that matched the number of channels of input data, which was 55. An LSTM layer with 100 hidden units was used. The amount of information learned by the layer was determined by the number of hidden units. While larger values may yield more accurate results, they can also increase susceptibility to overfitting to training data.

To specify the number of values to predict, a fully connected layer with a size matching the number of responses was included, followed by a regression layer. Here, there was a single response. The training options were specified as follows:-The training was performed using the Adam optimizer.-The network was trained for 100,000 epochs. For larger data sets, a lower number of epochs may suffice for achieving a good fit.-The sequences and responses used for validation were specified.-The learning rate was set at 0.0005.-The network that gave the best validation loss, i.e., the lowest validation loss, was outputted.

To evaluate the performance of the developed model in predicting the outputs of unknown inputs, five metabolic pathways were used as test data. After training the model, these test data were used for evaluation of the model output. The mentioned process was repeated 10 times; two of these are shown in Figure 13B as examples. Although the number of dataset instances was not high, the developed model showed its potential to predict output pattern and values. The calculated average R-squared value was about 0.64.

### 4.12. Utilizing AI and Genetic Algorithms to Identify Key Metabolites in the Metabolic Route

The integration of artificial intelligence (AI) in studying metabolic pathways has emerged as a valuable tool for identifying key metabolites that significantly influence diverse biological processes. The potential to discern these crucial metabolites can provide invaluable insights into various disease mechanisms, drug development, and therapeutic interventions. In the previous section of this research, a Long Short-Term Memory (LSTM) network-based model capable of processing 55 input metabolite sequences representing the metabolic routes was developed. The output merit of these metabolic routes was determined through a fuzzy logic model, which provided valuable context to our analysis. By searching through a 56-dimensional space, our objective was to optimize the model and pinpoint the most effective metabolites within the metabolic route. To achieve this, we leveraged the powerful genetic algorithm (GA) search and optimization technique, enabling us to efficiently explore the vast solution space. A comprehensive depiction of the entire process is presented in Figure 13C.

The strong optimization method known as the genetic algorithm (GA) is based on the concepts of natural selection and genetics. GAs have emerged as a popular and efficient technique for resolving challenging optimization issues in a variety of disciplines, including engineering, computer science, finance, and biology. By using a population of potential solutions and iteratively evolving them through selection, crossover, and mutation procedures, GAs simulate the process of evolution. GAs may effectively search huge solution spaces using this genetically inspired strategy, finding optimal or nearly optimal solutions to complex problems that would otherwise be difficult to solve using conventional optimization techniques. This section will delve into the fundamentals and uses of genetic algorithms, emphasizing their importance as a flexible and reliable approach for optimization [301,302]. To accomplish our ultimate goal of identifying the most influential metabolites in the metabolic route, we performed an optimization process. Our approach involved traversing a 56-dimensional space in which each dimension corresponded to the 55 metabolite inputs along with an additional dimension representing the output merit of the metabolic route. The optimization process was guided by a binary-GA search technique, which efficiently explored the solution space to find the optimal configuration of metabolite inputs that yielded the highest output merit [303]. The parameters that were used in developing the binary-GA search method for maximization of the model in the 56-dimensional space are indicated in Table 10.

## 5. Conclusions

In summary, we identified 30 genes, six miRNAs, and four lncRNAs that may play significant roles in nAMD pathogenesis. We also found three key metabolites that drive AMD development that are also common with AD and schizophrenia. Moreover, we identified nine key SNPs and their related genes with a critical role in nAMD pathogenesis and that are common with AD, schizophrenia, MS, and PD. These results will contribute to the development of diagnostic and therapeutic biomarkers for nAMD in the near future and could open new avenues to the design and/or repurposing of drugs related to novel targets. In addition, in a family with an AMD patient, screening other family members for SNPs may be useful in predicting their susceptibility to other neurodegenerative diseases. Obviously, if SNPs shared between AMD and other neurodegenerative diseases are identified in these family members, the necessity of controlling metabolic, transcriptomic, and proteomic profiles may also be beneficial.

This study also demonstrates the effectiveness of using artificial intelligence (specifically, an LSTM network), a fuzzy logic model, and a genetic algorithm to identify important metabolites in complex metabolic pathways. The 25 chosen metabolites had a significant impact on the output quality of the metabolic pathways. Our findings shed light on the underlying mechanisms governing diverse biological processes. This study contributed to the broader comprehension of metabolic regulation, which has significant implications for drug development and precision medicine, by focusing on the most effective metabolites. We outline a promising method for exploring and unraveling complex biological networks, establishing the groundwork for future advancements in the field of metabolic and disease pathway analysis.

## Figures and Tables

**Figure 1 pharmaceuticals-16-01555-f001:**
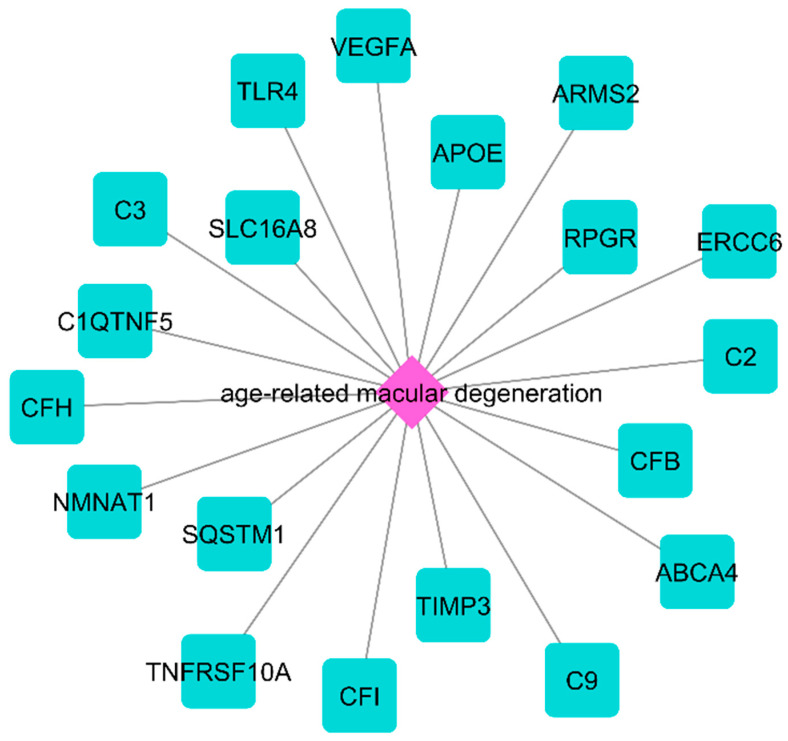
The proteins related to nAMD were identified using the NeDRex plugin. These nodes are displayed in blue due to the default settings of the plugin.

**Figure 2 pharmaceuticals-16-01555-f002:**
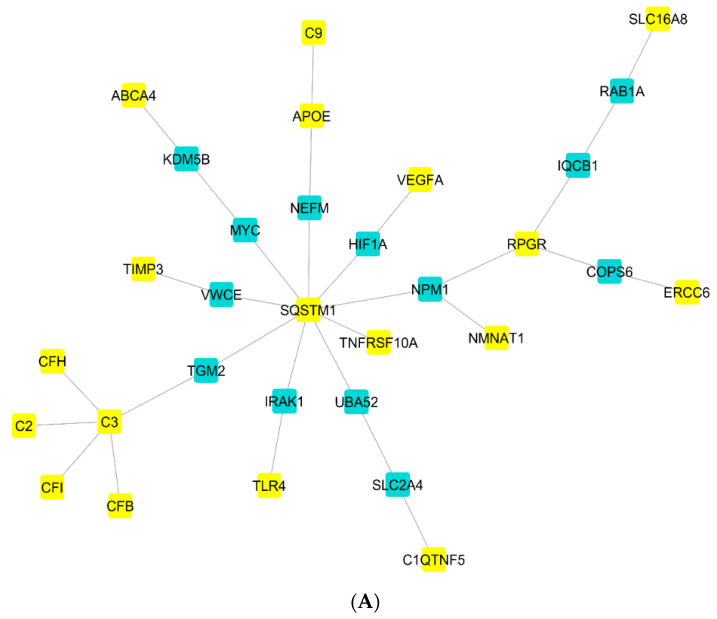
nAMD-related disease modules identification by two algorithms: (**A**) Multi-Steiner Trees (MuST) algorithm; (**B**) DIseAse MOdule Detection (DIAMOnD) algorithm. The blue nodes linked to AMD displayed in Figure 1 will change to yellow upon detecting disease modules due to the default settings of the plugin.

**Figure 3 pharmaceuticals-16-01555-f003:**
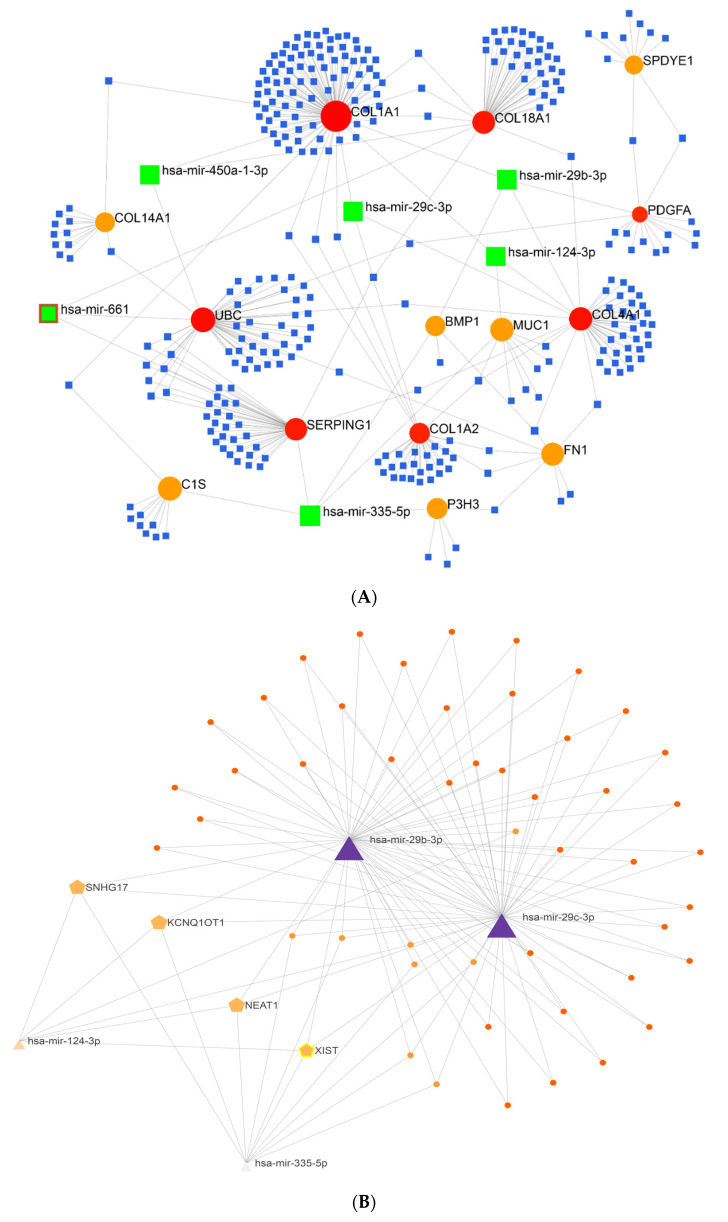
(**A**) The miRNA–gene regulatory network reconstruction identified six miRNAs and 14 essential genes by considering two centrality criteria: degree and betweenness. (**B**) Using six miRNAs as input data and the miRNet database (https://www.mirnet.ca (accessed on 18 August 2023)), a regulatory network of lncRNA-miRNA was reconstructed that identified four crucial lncRNAs. These four lncRNAs were deemed indispensable based on the two centrality criteria mentioned earlier.

**Figure 4 pharmaceuticals-16-01555-f004:**
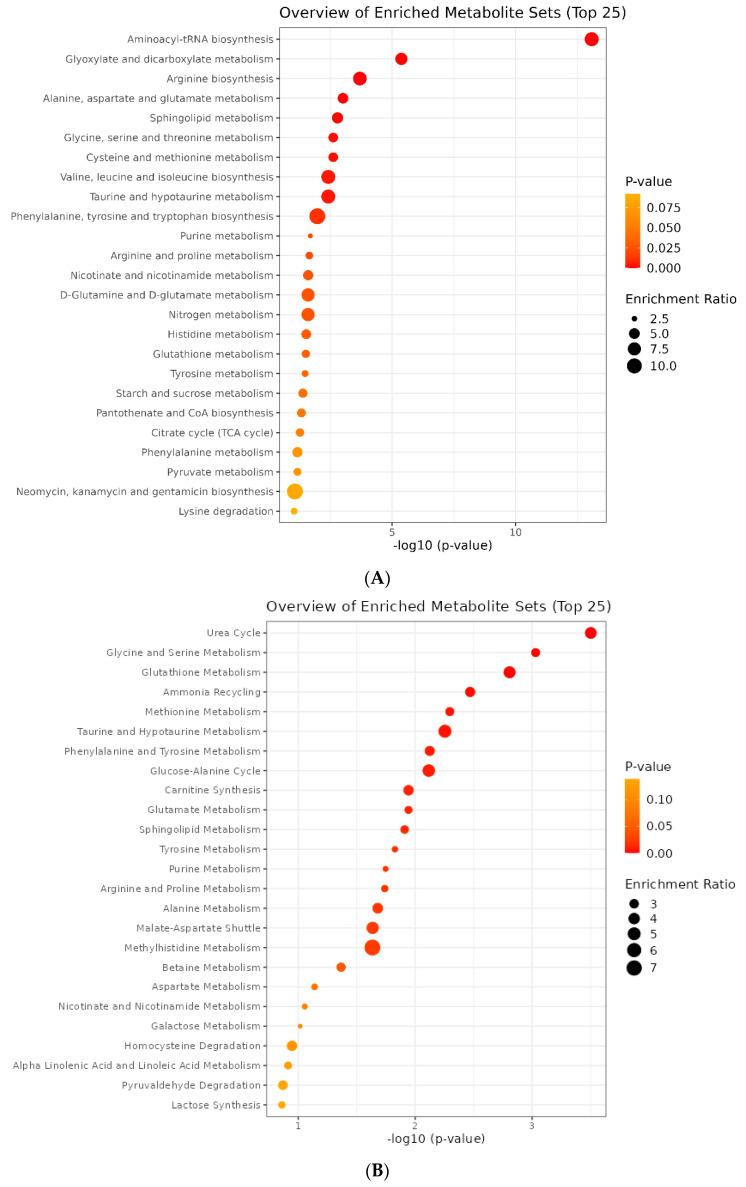
Metabolite enrichment analysis via different databases: (**A**) KEGG; (**B**) SMPDB.

**Figure 5 pharmaceuticals-16-01555-f005:**
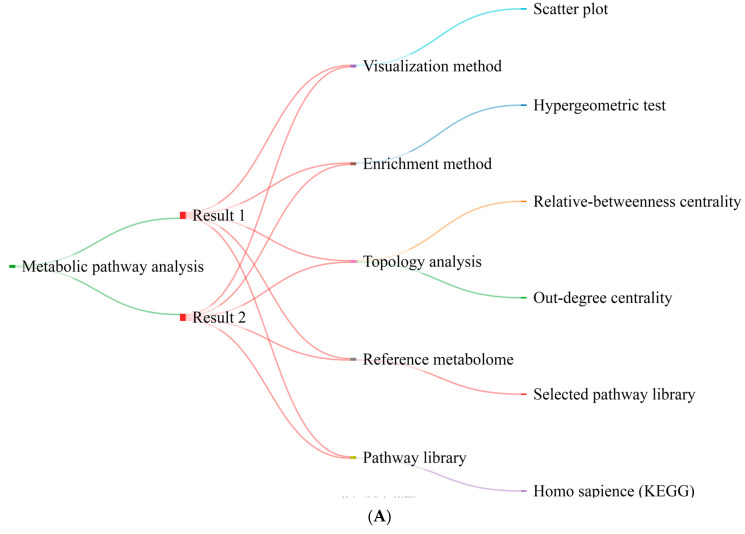
Metabolite-related analyses: (**A**) pathway analysis (result 1 with relative-betweenness centrality and result 2 via out-degree centrality); (**B**) joint pathway analysis (result 1 with degree centrality, result 2 via betweenness centrality, and result 3 via closeness centrality) with different criteria. Metabolic pathways that are repeatedly observed in various topological analyses are inherently more indispensable.

**Figure 6 pharmaceuticals-16-01555-f006:**
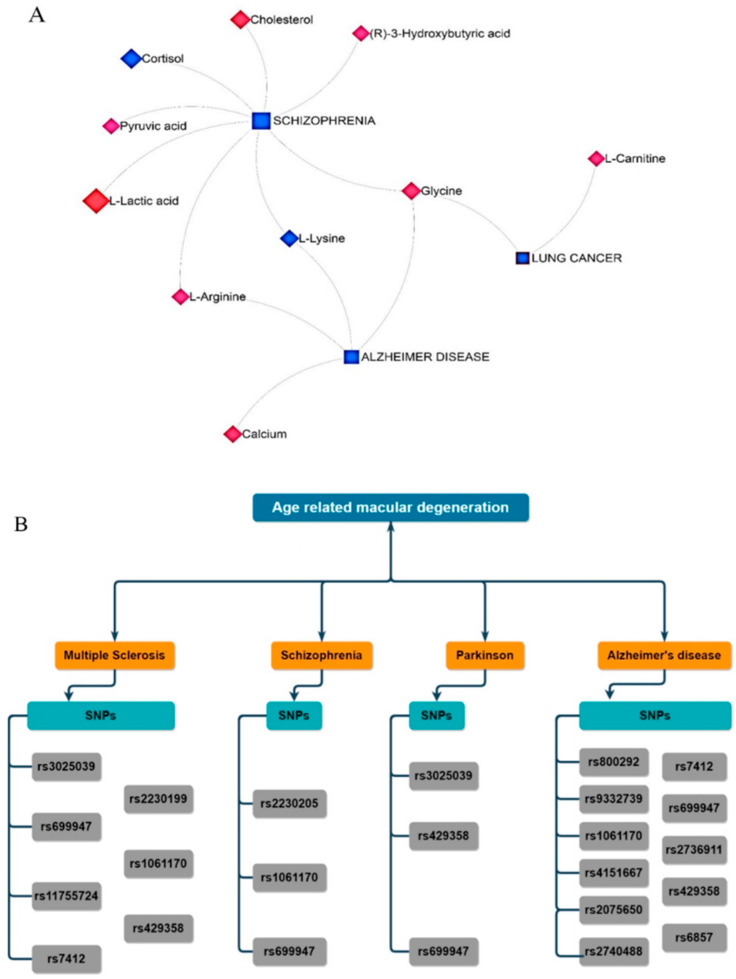
(**A**) Metabolite–gene–disease interaction network via different criteria. Three common metabolites (glycine, L-lysine, and L-arginine) were identified between AMD, Alzheimer’s disease, and schizophrenia. (**B**) Common SNPs between AMD and other neurodegenerative diseases, including Parkinson’s, Alzheimer’s, schizophrenia, and multiple sclerosis, were identified.

**Figure 7 pharmaceuticals-16-01555-f007:**
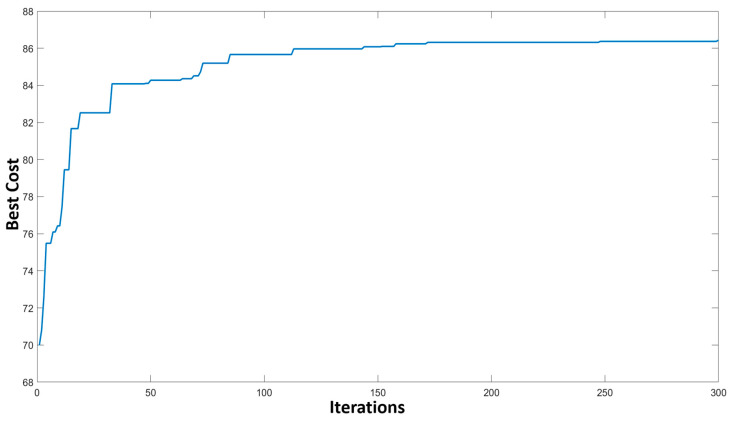
Maximization progress using 300 iterations of the binary-GA.

**Figure 8 pharmaceuticals-16-01555-f008:**
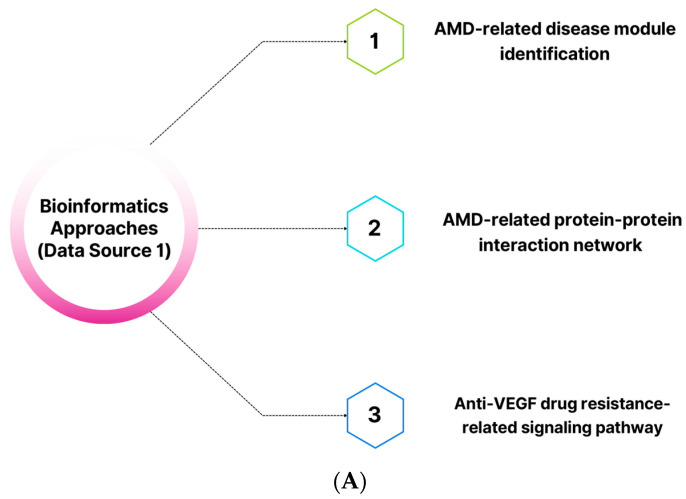
Tools used to collect data (genes). (**A**) Bioinformatics approaches designed for the Data Source 1 collection. This project utilized three general axes: 1. AMD-related disease module identification; 2. AMD-related protein–protein interactions; 3. The anti-VEGF drug resistance signaling pathway. (**B**) The process of identifying pathogenic modules associated with AMD disease involves several steps. Through the combination of the results from two algorithms (MuST and DIAMOnD), a collection of genes associated with the AMD disease modules was identified (gene list 1 = 230 genes). (**C**) The process and databases utilized to identify the proteins associated with AMD disease were designed and employed to construct the AMD-related protein–protein interaction network (gene list 2 = 7061 genes). (**D**) The third set of genes (4340 genes) are involved in developing resistance to anti-VEGF drugs. They were identified by analyzing 89 signaling pathways in the KEGG database and reviewing articles.

**Figure 9 pharmaceuticals-16-01555-f009:**
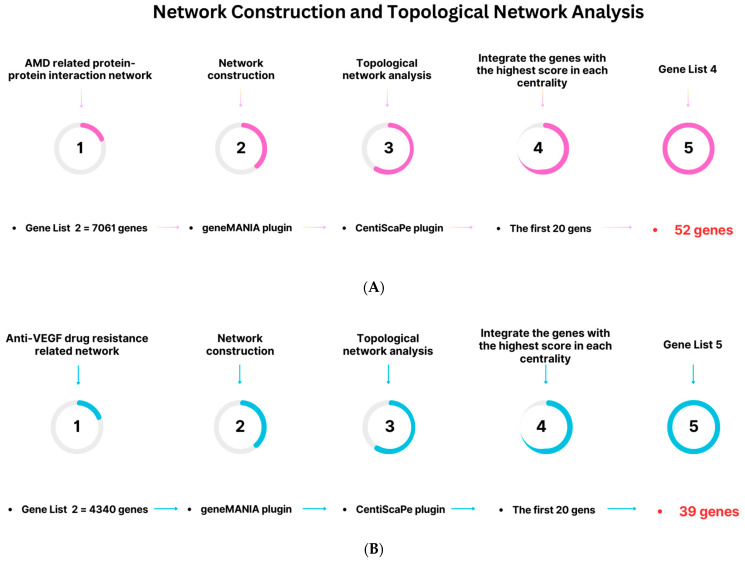
The schematic presentations show the sequential steps taken to construct the network and its topological analysis. (**A**) Data collection and topological network analysis were used to identify key proteins related to AMD or (**B**) anti-VEGF drug resistance. (**C**) The final gene list consisting of 313 genes was created by integrating data from the NeDRex platform, an AMD-related protein–protein interaction network, and information on drug resistance related to anti-VEGF treatment. (**D**) To identify the significant genes from the 313 identified genes, a topological network analysis was carried out again while taking into account 12 centrality parameters. The top 5 genes in each centrality were then chosen and combined to create a list of 31 genes, which is referred to as list number 7.

**Figure 10 pharmaceuticals-16-01555-f010:**
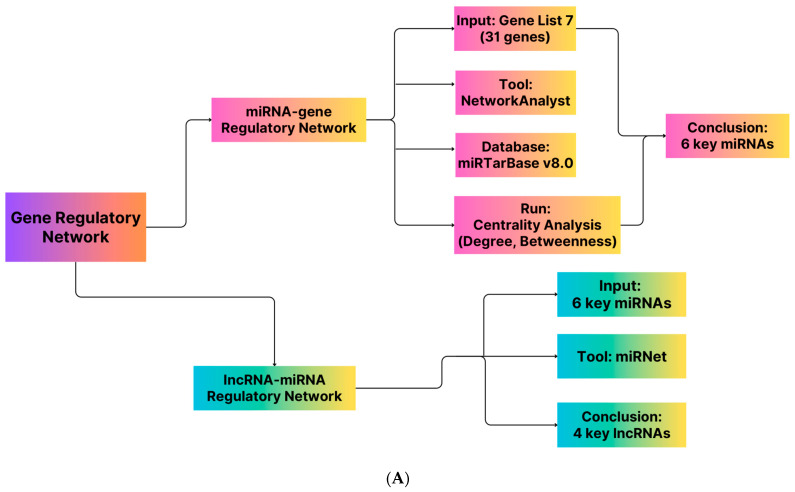
(**A**) A schematic presentation of the sequential steps taken and tools used to construct the gene regulatory network and its topological analysis. By considering the two parameters of centrality (degree and betweenness), 6 key miRNAs were identified through the gene–miRNA network analysis. In the following, 4 lncRNAs that were able to target all the miRNAs were also obtained through analyzing the lncRNA-miRNA network. (**B**) The tools used to collect nAMD-related data (metabolites and SNPs) and to perform the topological network analysis. By analyzing multiple networks, 10 key metabolites related to AMD and 30 critical SNPs related to AMD were identified by considering the two centrality parameters: degree and betweenness.

**Figure 11 pharmaceuticals-16-01555-f011:**
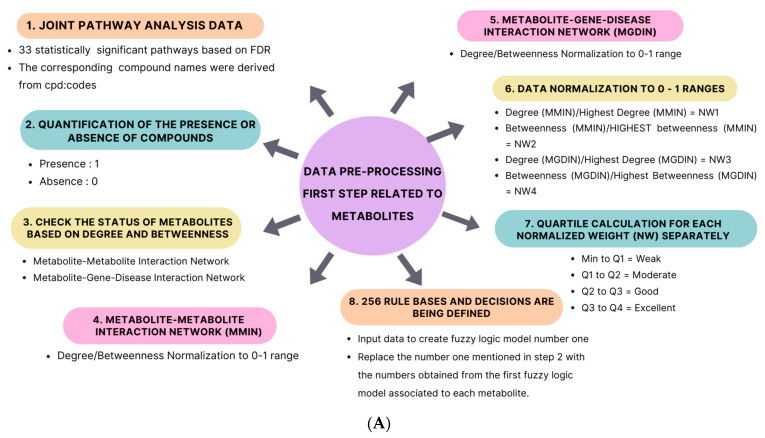
(**A**) In the first fuzzy logic model, the preliminary step of data pre-processing was performed on metabolites. To achieve this objective, a series of 8 steps was created. The specifics of each step are outlined in the diagram. (**B**) Metabolite merit calculations using the fuzzy logic model.

**Figure 12 pharmaceuticals-16-01555-f012:**
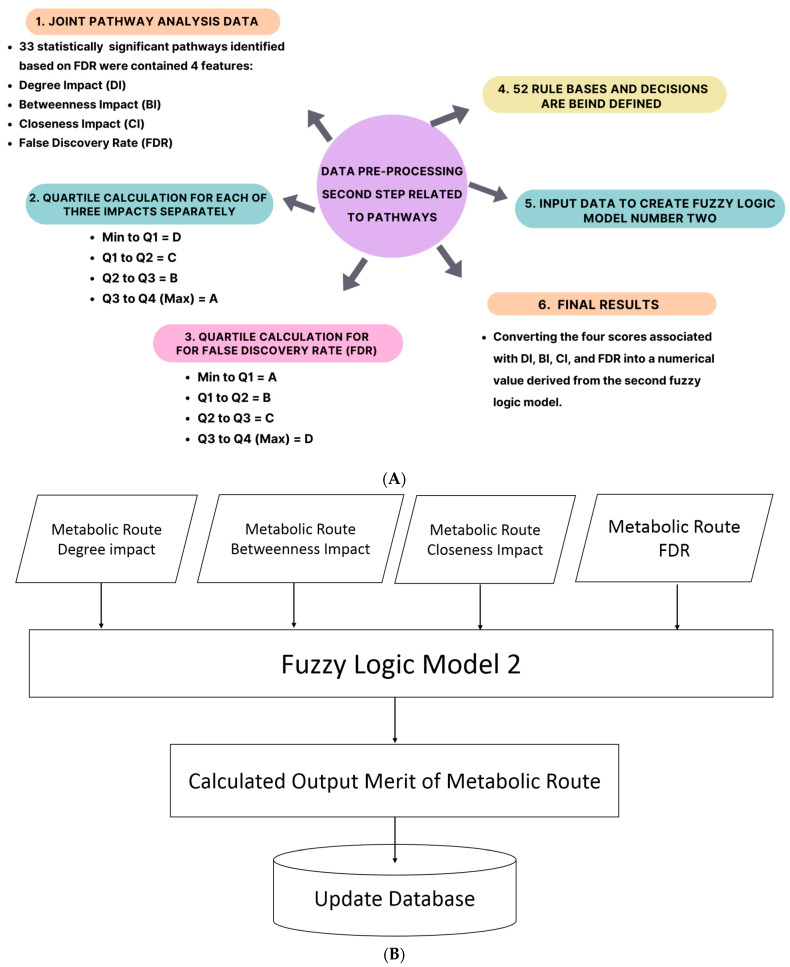
(**A**) In the second fuzzy logic model, the preliminary steps of data pre-processing were performed on metabolic pathways. To achieve this objective, a series of 8 steps was created. The specifics of each step are outlined in the diagram. (**B**) Output merit of metabolic route calculations using the fuzzy logic model.

**Figure 13 pharmaceuticals-16-01555-f013:**
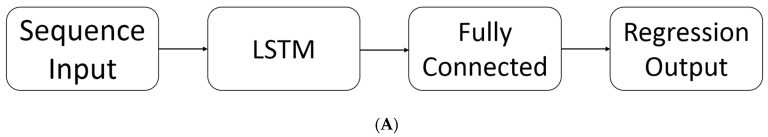
(**A**) General architecture of a Long Short-Term Memory (LSTM) network. (**B**) Two examples of test data classification using the developed LSTM network. (**C**) Comprehensive depiction of identifying the most effective metabolites in metabolic routes.

**Table 1 pharmaceuticals-16-01555-t001:** Metabolites derived via the fuzzy logic model, deep learning, and the genetic algorithm.

L-Leucine	Inosine	L-Glutamic Acid	L-Aspartate
Zinc (II) ion	SM(d18:1/18:0)	L-Alanine	L-Cystine
cis-Aconitic acid	Dopaquinone	L-Serine	L-Lysine
L-Aspartic acid	L-Histidine	Betaine	Adenosine
Glutathione	Cytidine	L-Arginine	L-Valine
Urea	Glycerol	L-Glutamine	Glycine
Taurine			

**Table 2 pharmaceuticals-16-01555-t002:** AMD-related results showing three significant gene lists, six miRNAs, four lncRNAs, and seven metabolites that were shared between the metabolite–metabolite interaction network (MMIN) and metabolite–gene–disease interaction network (MGDIN) via degree and betweenness centralities as well as 25 metabolites based on the fuzzy logic model, deep learning, and the genetic algorithm.

AMD-Related Results								
No.	Gene Name			miRNAs	lncRNAs	Metabolites		
	Disease Module	Gene Regulatory Network	AMD-SNP Data			Shared Metabolites between the MMIN and MGDIN Networks via Degree and Betweenness centralities	Fuzzy Logic Model + Deep Learning + Genetic Algorithm	
**1**	SLC16A8	PDGFA	CFH	hsa-mir-450a-1-3p	NEAT1	Pyruvic acid	L-Leucine	L-Glutamic acid
**2**	RPGR	COL1A1	VEGFA	hsa-mir-661	SNHG17	Glycine	Zinc (II) ion	L-Alanine
**3**	ERCC6	COL1A2	APOE	hsa-mir-335-5p	KCNQ1QT1	Citric acid	cis-Aconitic acid	L-Serine
**4**	NMNAT1	COL4A1	TOMM40	hsa-mir-124-3p	XIST	L-Lysine	L-Aspartic acid	Betaine
**5**	VEGFA	COL14A1	PVRL2	hsa-mir-29b-3p		L-Alanine	Glutathione	L-Arginine
**6**	TNFRSF10A	COL18A1	ABCA1	hsa-mir-29c-3p		L-Arginine	Urea	L-Glutamine
**7**	SQSTM1	UBC				L-Methionine	Taurine	L-Aspartate
**8**	C9	C1S					Inosine	L-Cystine
**9**	APOE	P3H3					SM(d18:1/18:0)	L-Lysine
**10**	TLR4	FN1					Dopaquinone	Adenosine
**11**	ABCA4	MUC1					L-Histidine	L-Valine
**12**	TIMP3	BMP1					Cytidine	Glycine
**13**	C3	SERPING1					Glycerol	
**14**	CFH	SPDYE1						
**15**	C2							
**16**	CFI							
**17**	CFB							
**18**	C1QTNF5							

**Table 3 pharmaceuticals-16-01555-t003:** Summary of the common metabolites between AMD, schizophrenia, and AD as well as the 30 AMD-SNPs and the shared SNPs between AMD and AD, MS, PD, and schizophrenia.

			AMD-Related Results					
No	Common Metabolites		AMD-SNPs			Common SNPs		
	AMD and Schizophrenia	AMD and Alzheimer’s Disease			AMD and Multiple Sclerosis	AMD and Schizophrenia	AMD and Parkinson’s Disease	AMD and Alzheimer’s Disease
1	L-Lactic acid	Glycine	rs17576	rs114254831	rs3025039	rs699947	rs3025039	rs800292
2	Cortisol	L-Lysine	rs1061170	rs116503776	rs699947	rs2230205	rs429358	rs9332739
3	Cholesterol	L-Arginine	rs699947	rs2740488	rs11755724	rs1061170	rs699947	rs1061170
4	(R)-3-Hydroxybutyric acid	Calcium	rs429358	rs12678919	rs7412			rs4151667
5	Glycine		rs2043085	rs7679	rs2230199			rs2075650
6	L-Lysine		rs3764261	rs3918242	rs1061170			rs2740488
7	L-Arginine		rs4073	rs800292	rs429358			rs7412
8	Pyruvic acid		rs243865	rs3025039				rs699947
9			rs964184	rs7412				rs2736911
10			rs2075650	rs2070895				rs429358
11			rs174547	rs1800775				rs6857
12			rs2071559	rs17577				
13			rs1800961	rs1065489				
14			rs6857	rs10468017				
15			rs1837253	rs17231506				

**Table 4 pharmaceuticals-16-01555-t004:** Findings of the pathway enrichment analysis, including information on 31 essential genes, six miRNAs, and 115 metabolites. The table also provides details on the pathway analysis and joint pathway analysis.

Pathway Enrichment Analysis						
NO	Genes	miRNAs	Metabolites			
			Enrichment Analysis		Metabolic Pathway Analysis	Joint Pathway Analysis
			KEGG	SMPDB		
**1**	Staphylococcus aureus infection (FDR = 0.000152)	Terpenoid backbone biosynthesis (FDR = 0.0022855)	Aminoacyl–tRNA biosynthesis (FDR = 7.02 × 10^−12^)	Urea cycle (FDR = 0.0308)	Aminoacyl–tRNA biosynthesis (FDR = 8.02 × 10^−12^)	Alanine, aspartate, and glutamate metabolism (FDR = 0.0005232)
**2**	Protein digestion and absorption (FDR = 0.00031)	Arachidonic acid metabolism (FDR = 0.0280283)	Glyoxylate and dicarboxylate metabolism (FDR = 0.000176)	Glycine and serine metabolism (FDR = 0.0458)	Alanine, aspartate, and glutamate metabolism (FDR = 0.021047)	Glycine, serine, and threonine metabolism (FDR = 0.000031504)
**3**	ECM–receptor interaction (FDR = 0.00427)	Hippo signaling pathway—multiple species (FDR = 0.0280283)	Arginine biosynthesis (FDR = 0.00562)		Glycine, Serine, and Threonine metabolism (FDR = 0.029711)	Arginine biosynthesis (FDR = 0.00083089)
**4**	Amoebiasis (FDR = 0.00492)	Base excision repair (FDR = 0.0311627)	Alanine, aspartate, and glutamate metabolism (FDR = 0.0204)		Taurine and Hypotaurine Metabolism (FDR = 0.03605)	Sphingolipid metabolism (FDR = 0.0072172)
**5**	AGE-RAGE signaling pathway in diabetes complications (FDR = 0.00492)	Complement and coagulation cascades (FDR = 0.0311627)	Sphingolipid metabolism (FDR = 0.027)			Cysteine and methionine metabolism (FDR = 0.000016837)
**6**	Focal adhesion (FDR = 0.00492)		Glycine, serine, and threonine metabolism (FDR = 0.0288)			Arginine biosynthesis (FDR = 0.000831)
**7**	Complement and coagulation cascades (FDR = 0.0401)		Cysteine and methionine metabolism (FDR = 0.0288)			
**8**			Valine, leucine, and isoleucine biosynthesis (FDR = 0.0354)			
**9**			Taurine and hypotaurine metabolism (FDR = 0.0354)			

**Table 5 pharmaceuticals-16-01555-t005:** Findings that were frequently seen in the metabolic pathway enrichment analysis, pathway analysis, and joint pathway analysis. The numbers shown in the final output section represent the frequency of pathway presence (+) in the outcomes of various analyses.

Pathways	Enrichment Analysis		Pathway Analysis Based on KEGG		Joint Pathway Analysis			Final Output
	KEGG	SMPDB	Relative Betweenness Centrality (R-b C)	Out-Degree Centrality (O-d C)	Degree	Betweenness	Closeness	
**Aminoacyl–tRNA Biosynthesis**	+	--	--	+	--	--	--	2
**Glyoxylate and dicarboxylate metabolism**	+	--	--	--	--	--	--	1
**Arginine biosynthesis**	+	--	--	--	+	--	+	3
**Alanine, aspartate, and glutamate metabolism**	+	--	+	+	+	+	--	5
**Sphingolipid metabolism**	+	--	--	--	+	--	+	3
**Glycine, serine, and threonine metabolism**	+	--	+	--	+	+	--	4
**Cysteine and methionine metabolism**	+	--	--	--	--	+	--	2
**Valine, leucine, and isoleucine biosynthesis**	+	--	--	--	--	--	--	1
**Taurine and hypotaurine metabolism**	+	--	+	+	--	--	--	3
**Urea cycle**	--	+	--	--	--	--	--	1
**Glycine and serine metabolism**	+	+	+	--	+	+	--	5

**Table 6 pharmaceuticals-16-01555-t006:** The range of four triangular membership functions.

	Weak	Moderate	Good	Excellent
**Normalized Weight 1**	−1 to 0.040625	0.040625 to 0.178125	0.178125 to 0.346875	0.346875 to 1
**Normalized Weight 2**	−1 to 0.005215122	0.005215122 to 0.035244476	0.035244476 to 0.114987363	0.114987363 to 1
**Normalized Weight 3**	−1 to 0.036363636	0.036363636 to 0.090909091	0.090909091 to 0.145454545	0.145454545 to 1
**Normalized Weight 4**	−1 to 0.005028148	0.005028148 to 0.046520343	0.046520343 To 0.093742947	0.093742947 to 1

**Table 7 pharmaceuticals-16-01555-t007:** The calculated merits for each metabolite.

No.	Metabolites	Normalized Weight 1	Normalized Weight 2	Normalized Weight 3	Normalized Weight 4	Calculated Merits
**1**	Maltotriose	0.059375	0.012879969	−1	−1	12.33015695
**2**	L-Glutamic acid	0.95625	0.892826812	0.090909091	0.043457267	42.4297409
**3**	Pyruvic acid	0.778125	0.558492981	0.272727273	0.088754688	88.06311429
**4**	L-Tryptophan	0.296875	0.107651537	0.109090909	0.109603133	62.5
**5**	Citric acid	0.45	0.170635833	0.181818182	0.137017563	90.33333333
**6**	L-Alanine	0.39375	0.112928955	0.218181818	0.093742947	87.32827717
**7**	L-Serine	0.36875	0.084920841	0.127272727	0.045147312	62.5
**8**	Betaine	0.15625	0.027372191	0.090909091	0.048277571	42.63825352
**9**	Dimethyl sulfone	0.021875	0.000196837	−1	−1	11.6245581
**10**	L-Arginine	0.371875	0.131651201	0.290909091	0.14684817	89.35621737
**11**	Sphinganine	0.06875	0.015722582	−1	−1	11.6568779
**12**	L-Glutamine	0.371875	0.075654603	0.109090909	0.047897573	62.5
**13**	L-Tyrosine	0.25	0.068912558	0.090909091	0.037760448	42.4297409
**14**	Cholesterol sulfate	0.003125	0	0.418181818	0.475029838	90.25525526
**15**	Sucrose	0.178125	0.06483723	0.018181818	0	12.90686029
**16**	L-Phenylalanine	0.28125	0.049343156	0.109090909	0.091631965	62.5
**17**	L-Cysteine	0.4125	0.143670141	0.090909091	0.080663464	62.5
**18**	L-Aspartate-semialdehyde	0.040625	5.06488 × 10^−5^	−1	−1	13.82051282
**19**	L-Methionine	0.334375	0.075768988	0.181818182	0.20829445	86.56442358
**20**	Creatine	0.1	0.016946398	0.090909091	0.070195678	42.4297409
**21**	L-Cystine	0.00625	0	0.127272727	0.041291699	62.5
**22**	L-Lysine	0.403125	0.136364416	0.272727273	0.147600818	90.33333333
**23**	L-Isoleucine	0.2	0.009885279	0.127272727	0.046520343	44.40997593
**24**	Phytosphingosine	0.034375	0.005215122	−1	−1	13.61538462
**25**	Adenosine	0.346875	0.119206393	0.054545455	0.0280831	37.5
**26**	L-Valine	0.2375	0.020285683	0.145454545	0.068937066	62.5
**27**	Glycine	0.4875	0.217186109	0.363636364	0.259194216	90.33333333
**28**	L-Leucine	0.28125	0.039149797	0.145454545	0.080064075	62.5
**29**	cis-Aconitic acid	0.121875	0.018639401	0.054545455	0.01249794	37.5
**30**	Hypotaurine	0.03125	0.000358991	−1	−1	13.08960132
**31**	S-Adenosylhomocysteine	0.571875	0.575112844	0.036363636	0.005028148	37.5
**32**	L-Aspartic acid	0.359375	0.114987363	0.054545455	0.013255836	37.5
**33**	Glutathione	0.296875	0.105460029	0.072727273	0.085964538	62.5
**34**	Arachidonic acid	0.228125	0.167812851	0.036363636	0.027691555	37.5
**35**	Adenine	0.3	0.12548383	−1	−1	11.49545672
**36**	Urea	0.153125	0.059641216	0.090909091	0.011850264	42.4297409
**37**	Serotonin	0.3375	0.226082231	0.072727273	0.062262961	54.87490171
**38**	Taurine	0.153125	0.047879708	0.090909091	0.077994031	62.5
**39**	L-Lactic acid	0.1	0.013350728	1	1	89.44287908
**40**	p-Hydroxyphenylacetic acid	0.028125	0.000479396	0.054545455	0.035107811	37.5
**41**	Inosine	0.153125	0.021684479	0.054545455	0.029713312	37.5
**42**	SM(d18:1/18:0)	0.040625	0.014353102	−1	−1	13.82051282
**43**	Hypoxanthine	0.175	0.029327862	0.090909091	0.108400157	62.5
**44**	Dopaquinone	0.01875	0.000234529	−1	−1	11.19413764
**45**	L-Proline	0.2125	0.042078003	0.2	0.10544835	87.16355188
**46**	L-Histidine	0.23125	0.035244476	0.127272727	0.048363647	53.99094217
**47**	Guanine	0.165625	0.023106832	0.018181818	0	11.96511859
**48**	Cytidine	0.134375	0.02976407	0.018181818	0	11.41056611
**49**	Glycerol	1	1	0.109090909	0.169397158	89.55878284
**50**	Calcium	−1	−1	0.327272727	0.318269183	90.33333333
**51**	Acetoacetic acid	−1	−1	0.127272727	0.064575489	62.5
**52**	Formic acid	−1	−1	0.018181818	0	11.41056611
**53**	Zinc (II) ion	−1	−1	0.072727273	0.025154387	37.5
**54**	Acetic acid	−1	−1	0.054545455	0.012651199	37.5
**55**	Cortisol	−1	−1	0.418181818	0.65988514	90.33333333

**Table 8 pharmaceuticals-16-01555-t008:** The range of four independent parameters.

	D	C	B	A
**Degree Impact**	0 to 0.044118	0.044118 to 0.11111	0.11111 to 0.213095	0.213095 to 0.4918
**Betweenness Impact**	0 to 0	0 to 0	0 to 0.02748	0.02748 to 0.17737
**Closeness Impact**	0.062116 to 0.0994065	0.0994065 to 0.13354	0.13354 to 0.21584	0.21584 to 0.67273
	**A**	**B**	**C**	**D**
**FDR**	4.3423 × 10^−27^ to 2.41705 × 10^−5^	2.41705 × 10^−5^ to 0.0092785	0.0092785 to 0.0282315	0.0282315 to 0.046353

**Table 9 pharmaceuticals-16-01555-t009:** The calculated merits for each pathway.

No.	Metabolic Pathway	Degree Impact	Betweenness Impact	Closeness Impact	FDR	Calculated Merits
**1**	ABC transporters	0	0	0.2	4.3423 × 10^−27^	90.04878049
**2**	Protein digestion and absorption	0	0	0.26027	5.1706 × 10^−26^	90.04878049
**3**	Central carbon metabolism in cancer	0	0	0.15789	5.2636 × 10^−20^	88.17843178
**4**	Aminoacyl–tRNA biosynthesis	0.18557	0	0.1887	3.0666 × 10^−16^	89.04243328
**5**	Mineral absorption	0.058824	0	0.16361	7.8109 × 10^−13^	88.29666352
**6**	Glyoxylate and dicarboxylate metabolism	0.19318	0.005094	0.14279	2.6159 × 10^−7^	65
**7**	Taurine and hypotaurine metabolism	0.27586	0	0.26261	4.1941 × 10^−7^	90.04862
**8**	Cysteine and methionine metabolism	0.23301	0.17737	0.17003	0.000016837	88.72511315
**9**	Glycine, serine, and threonine metabolism	0.37647	0.14892	0.23869	0.000031504	90.03666656
**10**	Alanine, aspartate, and glutamate metabolism	0.4918	0.12764	0.1311	0.0005232	88.6812472
**11**	Ferroptosis	0.094595	0	0.16278	0.00080319	50
**12**	Sulfur metabolism	0	0	0.10345	0.00080319	36.6802727
**13**	Arginine biosynthesis	0.37143	0.067227	0.42256	0.00083089	89.70463799
**14**	Amoebiasis	0.016667	0	0.13142	0.0012241	36.7479835
**15**	Valine, leucine, and isoleucine biosynthesis	0.15385	0	0.15685	0.0016043	88
**16**	Sphingolipid metabolism	0.34426	0.11199	0.67273	0.0072172	89.00877813
**17**	Phenylalanine metabolism	0.11111	0.011281	0.076586	0.0092785	43.91079969
**18**	Purine metabolism	0.16832	0.014208	0.11081	0.010746	66.68460826
**19**	Thiamine metabolism	0.026316	0	0.11902	0.010746	36.7661384
**20**	Pantothenate and CoA biosynthesis	0.095238	0	0.1303	0.010746	36.78520447
**21**	Arginine and proline metabolism	0.1453	0.039309	0.10489	0.01168	65
**22**	Staphylococcus aureus infection	0.093023	0.0011074	0.095363	0.021438	36.83966667
**23**	Gap junction	0.076923	0	0.24965	0.021454	50
**24**	Neuroactive ligand–receptor interaction	0.061798	0	0.062116	0.023787	30.30921831
**25**	Primary bile acid biosynthesis	0.09375	0	0.23168	0.027448	50
**26**	AGE-RAGE signaling pathway in diabetic complications	0.029412	0	0.13354	0.029015	36.71982118
**27**	Tyrosine metabolism	0.1129	0.04149	0.094005	0.032855	65
**28**	Butanoate metabolism	0.16364	0.016498	0.089484	0.032855	88
**29**	Pyruvate metabolism	0.33962	0.038462	0.25425	0.032855	88.14549064
**30**	Taste transduction	0.14085	0	0.12007	0.032961	36.73783459
**31**	Nitrogen metabolism	0.065217	0	0.069877	0.036419	14.3852384
**32**	Carbohydrate digestion and absorption	0	0	0.080645	0.036419	13.94718423
**33**	Phenylalanine, tyrosine, and tryptophan biosynthesis	0.29268	0.0093496	0.085978	0.046353	50

**Table 10 pharmaceuticals-16-01555-t010:** The parameters that were used in the developed binary-GA search method for maximization of the model in the 56-dimensional space.

Maximum iteration	300
Population size	100
Number of chromosomes	55
Mutation coefficient	0.09
Proportion of crossover	1

## Data Availability

Data is contained within the article and Appendix A.

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
