# Peer review of "Construction of an Exudative Age-Related Macular Degeneration Diagnostic and Therapeutic Molecular Network Using Multi-Layer Network Analysis, a Fuzzy Logic Model, and Deep Learning Techniques: Are Retinal and Brain Neurodegenerative Disorders Related?"

_pharmaceuticals, 2023, doi:10.3390/ph16111555_

Round 1
Reviewer 1 Report
Comments and Suggestions for Authors
Review of a manuscript “Construction of an exudative AMD diagnostic and therapeutic molecular network using multi-layer network analysis, fuzzy logic model, and deep learning techniques: Are retinal and brain neurodegenerative disorders related?” By Hamid Latifi-Navid and coauthors submitted to “Pharmaceuticals”
Neovascular age-related macular degeneration (nAMD) is a primary cause of irreversible visual impairment in bringing enormous problems for patients, their relatives and health care system worldwide. Several approaches might be used to reduce VEGF-VEGFR2 interactions, including antibody-based methods and tyrosine kinase inhibitory systems, however the mechanism of their action should be examined more thoroughly. The authors aimed to identify disease-related modules associated with nAMD and to study how various angiogenesis signaling pathways are involved in the pathogenesis. This is an important area of investigation in vision research and the results will be interesting for the readers of “Pharmaceuticals”
The following corrections and additions should be made.
Abstract
The majority of space in Abstract is occupied by enumeration of the objective of the study (seven of them are listed). After listing these aims, the authors switch to results demonstrate the key players participating in pathophysiology of nAMD without giving enough materials concerning the results.The Abstract should be rewritten giving a more balanced content, emphasizing the most important results of the study.
Introduction
Lines 60-61:” These include the compensatory angiogenic pathways, vessel cooption, intussusceptive microvascular growth, and vascular mimicry”. Please, add references after this sentence.
Lines 55-57:”Despite promising results with anti-VEGF monotherapies in different neovascular diseases, a plethora of current studies reveal an outbreak of resistance and/or lack of response to anti-angiogenic drugs including anti-VEGF in significant portion of nAMD patients.” References should be added here.
Results
Figure 1. The nAMD-related proteins were identified through NeDRex plugin. It is unclear why blue highlighters mark only parts of the symbols on this figure.
Then same concerns Figure 2.
Lines 144-145
“…and 14 essential genes were found…” It is unclear how they were found. Please, explain why these genes were selected.
Figure 4.
The text left from the figure is blurred and hard to read.
Line 207. “We simultaneously analyzed 115 unique metabolites” Please, explain why these 115 metabolite were selected.
Figure 7. Maximization progress using 300 iterations of the Binary-GA. Please, increase the fonts in the text for easier reading. The same may concern other figures.
3.5. Autophagy and AMD.
Lines 631-632: ”Mitochondrial dysfunction, mtDNA damage and increased ROS production 631 lead to protein aggregation and inflammation in AMD [149].” Please, add the following reference after this sentence: ”Protein aggregation in retinal cells and approaches to cell protection. Cell Mol Neurobiol. 2005 Sep;25(6):1051-66. doi: 10.1007/s10571-005-8474-1.”
3.6. None-coding RNAs and AMD. This section of the manuscript is too long and should be made more concise.
Conclusion
Line 1167:”In summary, we identified thirty genes, six miRNAs, and four lncRNAs with critical roles in nAMD pathogenesis.” Less categorical statement would be beneficial here, like “In summary, we identified thirty genes, six miRNAs, and four lncRNAs which may play significant roles in nAMD pathogenesis.”
Author Response
Reviewer #1
Point #1I
Abstract
The majority of space in Abstract is occupied by enumeration of the objective of the study (seven of them are listed). After listing these aims, the authors switch to results demonstrate the key players participating in pathophysiology of nAMD without giving enough materials concerning the results.The Abstract should be rewritten giving a more balanced content, emphasizing the most important results of the study. The abstract is now revised as suggested.
Point #2I
Introduction
Lines 60-61:” These include the compensatory angiogenic pathways, vessel cooption, intussusceptive microvascular growth, and vascular mimicry”. Please, add references after this sentence. We have now added appropriate references.
Point #3I
Lines 55-57:”Despite promising results with anti-VEGF monotherapies in different neovascular diseases, a plethora of current studies reveal an outbreak of resistance and/or lack of response to anti-angiogenic drugs including anti-VEGF in significant portion of nAMD patients.” References should be added here. We have now included appropriate references.
Point #4I
Figure 1. The nAMD-related proteins were identified through NeDRex plugin. It is unclear why blue highlighters mark only parts of the symbols on this figure. Then same concerns Figure 2.
We did not select the blue and yellow colors seen in Figures 1 and 2, they are rather the default colors of the NeDRex plugin. The nodes related to AMD disease in Figure 1 appear in blue, but when identifying disease modules in Figure 2, they are changed to yellow due to the plugin's default settings. This is now explained in the main text.
Point #5I
Lines 144-145
“…and 14 essential genes were found…” It is unclear how they were found. Please, explain why these genes were selected. The 14 essential genes identified at this stage are the same genes related to miRNAs in the figure showing gene regulatory network (Figure 3A). An explanation as how these 14 genes were identified is also given in the main text in order to help the reader for more clarity.
Point #6I
Figure 4. The text left from the figure is blurred and hard to read. We have revised the figure and high-quality images are provided.
Point #7I
Line 207. “We simultaneously analyzed 115 unique metabolites” Please, explain why these 115 metabolite were selected. As described in the materials and methods section (“4.5. Second data sources: nAMD-related metabolites and SNPs”), the 115 metabolites were gathered from the review of published articles that explore the connection between metabolites and the AMD disease. The references that were utilized for gathering the data are as follows.
-Li, X.; Cai, S.; He, Z.; Reilly, J.; Zeng, Z.; Strang, N.; Shu, X. Metabolomics in Retinal Diseases: An Update. Biology (Basel) 2021, 10, doi:10.3390/biology10100944.
-Brown, C.N.; Green, B.D.; Thompson, R.B.; den Hollander, A.I.; Lengyel, I.; consortium, E.-R. Metabolomics and Age-Related Macular Degeneration. Metabolites 2018, 9, doi:10.3390/metabo9010004.
-Hou, X.W.; Wang, Y.; Pan, C.W. Metabolomics in Age-Related Macular Degeneration: A Systematic Review. Invest Ophthalmol Vis Sci 2020, 61, 13, doi:10.1167/iovs.61.14.13.
-Lains, I.; Duarte, D.; Barros, A.S.; Martins, A.S.; Gil, J.; Miller, J.B.; Marques, M.; Mesquita, T.; Kim, I.K.; Cachulo, M.D.L.; et al. Human plasma metabolomics in age-related macular degeneration (AMD) using nuclear magnetic resonance spectroscopy. PLoS One 2017, 12, e0177749, doi:10.1371/journal.pone.0177749.
Point #8I
Figure 7. Maximization progress using 300 iterations of the Binary-GA. Please, increase the fonts in the text for easier reading. The same may concern other figures. The figure is revised as suggested.
Point #9I
3.5. Autophagy and AMD.
Lines 631-632: ”Mitochondrial dysfunction, mtDNA damage and increased ROS production 631 lead to protein aggregation and inflammation in AMD [149].” Please, add the following reference after this sentence: ”Protein aggregation in retinal cells and approaches to cell protection. Cell Mol Neurobiol. 2005 Sep;25(6):1051-66. doi: 10.1007/s10571-005-8474-1.” This reference is now added.
Point #10I
3.6. None-coding RNAs and AMD. This section of the manuscript is too long and should be made more concise. This is now addressed as suggested.
Point #11I
Conclusion
Line 1167:”In summary, we identified thirty genes, six miRNAs, and four lncRNAs with critical roles in nAMD pathogenesis.” Less categorical statement would be beneficial here, like “In summary, we identified thirty genes, six miRNAs, and four lncRNAs which may play significant roles in nAMD pathogenesis.” This is now corrected as suggested.
Reviewer 2 Report
Comments and Suggestions for Authors
Thank you for the opportunity to review this manuscript entitled ‘Construction of an exudative AMD diagnostic and therapeutic molecular network using multi-layer network analysis, fuzzy logic model, and deep learning techniques: Are retinal and brain neurodegenerative disorders related?’ by Hamid Latifi-Navid and Colleagues. The authors used an integrated approach to summarize knowledge about the mechanisms of pathogenesis of exudative AMD using modern technologies. The authors obtained important information about the involvement of a number of genes, single nucleotide polymorphisms and metabolites in the mechanism of AMD development. This study is relevant due to the increasing incidence of AMD and the increasing social burden on the healthcare system in connection with this. The manuscript is presents data appropriate for publication in this journal. However, there are some key points that require clarification.
I recommend that the authors rewrite the abstract so that it matches the title of the manuscript. In the presented version there is no answer to the question that the authors themselves ask: Are retinal and brain neurodegenerative disorders related?
Please note that the references you provide in the ‘Introduction’ section are not included in the references list, which begins with the works you refer to in the ‘Discussion’ section.
The ‘Discussion’ section is overloaded with information and should be shortened. This note also applies to the ‘Methods’ section.
Captions to figures and tables require additional information.
For example: Figure 3. (A) miRNA-gene regulatory network reconstruction identified six miRNAs and 14 essential genes discovered by analyzing AV-DRN and AMD-PPIN
or Figure 4. Metabolite enrichment analysis of nAMD-related data via different databases. (A) KEGG and ( B) SMPDB.
Please check the captions for all figures.
In supplementary file the tables numbered incorrectly. Make the appropriate changes.
Figure 8B-C duplicates the information the supplementary Table 3. They need to be removed.
Specify the meaning of the colors in the supplementary table cells.
Author Response
Reviewer #2
Point #1 recommend that the authors rewrite the abstract so that it matches the title of the manuscript. In the presented version there is no answer to the question that the authors themselves ask: Are retinal and brain neurodegenerative disorders related? The abstract is now revised as suggested.
Point #2 Please note that the references you provide in the ‘Introduction’ section are not included in the references list, which begins with the works you refer to in the ‘Discussion’ section. This is now corrected.
Point #3 The ‘Discussion’ section is overloaded with information and should be shortened. This note also applies to the ‘Methods’ section. The suggested correction was performed in the discussion. Due to the complexity of the methods used and the need to maintain coherence of the content, great effort was made to convey concepts in a clear and concise manner. The structure of the methods section is crucial, and any changes could limit the scientific understanding of what was performed. However, to improve the article's quality, some parts were removed from the discussion section, and quality of all images were significantly improved.
Point #4 Captions to figures and tables require additional information. For example: Figure 3. (A) miRNA-gene regulatory network reconstruction identified six miRNAs and 14 essential genes discovered by analyzing AV-DRN and AMD-PPIN or Figure 4. Metabolite enrichment analysis of nAMD-related data via different databases. (A) KEGG and ( B) SMPDB. Please check the captions for all figures. The figure legends have been improved as suggested.
Point #5 In supplementary file the tables numbered incorrectly. Make the appropriate changes.
The correspondence of the numbers of supplementary tables with the main text was re-evaluated and corrected.
Point #6 Figure 8B-C duplicates the information in the supplementary Table 3. They need to be removed. The requested corrections were performed.
Point #7 Specify the meaning of the colors in the supplementary table cells. The colors used do not have any specific significance. They were simply used to ensure that the process was carried out correctly and without any errors during the sorting and utilization of data in the fuzzy logic and deep learning section. All the colors are removed in the revised document.
Reviewer 3 Report
Comments and Suggestions for Authors
Thanks for offering the opportunity to review this paper. There are certain issues with this paper that need to be addressed, so I would like to review it again after the author carefully modifies it.
1. Minor editing of English language required. On the whole, I can understand what the authors are trying to express in this manuscript, but there are some places where the expression is not smooth. In addition, there are some grammatical errors or clerical errors in the manuscript.
2.Tables need improvement.
3. The Figures are too blurry, so it is necessary to improve the resolution.
Comments on the Quality of English Language
Minor editing of English language required
Author Response
Point #1 Minor editing of English language required. On the whole, I can understand what the authors are trying to express in this manuscript, but there are some places where the expression is not smooth. In addition, there are some grammatical errors or clerical errors in the manuscript. We asked two native English-speaking scientists to read the manuscript and edit accordingly as indicated.
Point #2 Tables need improvement. Due to the complexity of the methods and results in this manuscript and the need to maintain coherence of the content, a great effort was made to convey concepts in a clear, and concise manner. The structure of the methods and results section is crucial, and any changes could lead to a disturbance in scientific understanding of the work presented.
Point #3 The Figures are too blurry, so it is necessary to improve the resolution. The images were either enhanced in quality and resolution or redrawn in certain instances to improve as suggested.
Round 2
Reviewer 2 Report
Comments and Suggestions for Authors
Thank you for the opportunity to review new version manuscript. The authors successfully edited the abstract of the manuscript and the signatures under some figures. Нowever, they ignored the comments regarding the change in the names of tables No. 2-5. The title of the table should reflect its content. This will improve your perception of the your results. In addition, the authors did not shorten the discussion section. I recommend making these changes to the manuscript.
Author Response
We have addressed the concerns of the reviewer as requested.